# Isoform specific anti-TGFβ therapy enhances antitumor efficacy in mouse models of cancer

Aditi Gupta[1,2], Sadna Budhu [1,2], Kelly Fitzgerald[1,2], Rachel Giese[1,2], Adam O. Michel[3], Aliya Holland[1,2], Luis Felipe Campesato[1,2], Jacques van Snick[4], Catherine Uyttenhove[4], Gerd Ritter [5], Jedd D. Wolchok [1,2,6,7✉] & Taha Merghoub [1,2,6,7✉]

TGFβ is a potential target in cancer treatment due to its dual role in tumorigenesis and homeostasis. However, the expression of TGFβ and its inhibition within the tumor micro-environment has mainly been investigated in stroma-heavy tumors. Using B16 mouse melanoma and CT26 colon carcinoma as models of stroma-poor tumors, we demonstrate that myeloid/dendritic cells are the main sources of TGFβ1 and TGFβ3. Depending on local expression of TGFβ isoforms, isoform specific inhibition of either TGFβ1 or TGFβ3 may be effective. The TGFβ signature of CT26 colon carcinoma is defined by TGFβ1 and TGFβ1 inhibition results in tumor delay; B16 melanoma has equal expression of both isoforms and inhibition of either TGFβ1 or TGFβ3 controls tumor growth. Using T cell functional assays, we show that the mechanism of tumor delay is through and dependent on enhanced CD8+ T cell function. To overcome the local immunosuppressive environment, we found that combining TGFβ inhibition with immune checkpoint blockade results in improved tumor control. Our data suggest that TGFβ inhibition in stroma poor tumors shifts the local immune environment to favor tumor suppression.

[1] Swim Across America and Ludwig Collaborative Laboratory, Immunology Program, Parker Institute for Cancer Immunotherapy, Memorial Sloan Kettering Cancer Center, New York, NY 10065, USA. [2] Human Oncology & Pathogenesis Program, Memorial Sloan Kettering Cancer Center, New York, NY 10065, USA. [3] Laboratory of Comparative Pathology, Center of Comparative Medicine and Pathology, Memorial Sloan Kettering Cancer Center, New York, NY 10065, USA. [4] Ludwig Institute for Cancer Research Ltd, Brussels, Belgium. [5] Ludwig Institute for Cancer Research Ltd, New York, NY, USA. [6] Department of Medicine, Memorial Sloan Kettering Cancer Center, New York, NY, USA. [7] Weill Cornell Medical College, New York, NY 10065, USA. ✉email: wolchokj@mskcc.org; merghout@mskcc.org

Transforming growth factor-β (TGFβ) is part of a complex signaling pathway due to its dichotomous roles in normal tissue homeostasis and carcinogenesis[1]. As a pleiotropic cytokine, TGFβ is involved in regulating cell growth, differentiation, motility, apoptosis, angiogenesis, and immune responses[1,2]. Depending upon the local context, TGFβ can function as either a tumor suppressor or a tumor promoter[1,3–6]. In a pre-malignant state, TGFβ is thought to inhibit tumor growth by limiting proliferation and inducing apoptosis. Certain cancer types, such as colorectal cancer (CRC), hepatocellular carcinoma, and lung cancer, are able to circumvent the cytostatic and apoptotic effects of TGFβ by mutating key players in its signaling cascade, allowing the transformed cells to undergo unrestrained growth[3,4,7,8]. However, other tumor types, including melanoma, glioma, and breast cancer, maintain intact TGFβ signaling but become less responsive to TGFβ-mediated growth suppression through the acquisition of compound oncogenic mutations[4,8]. These cancers then commandeer the canonical TGFβ pathway to promote epithelial-to-mesenchymal transition (EMT), modulate the extracellular environment (ECM), and decrease immune surveillance, leading to metastasis and treatment resistance[3,4,8–12].

Further layers of complication are derived from the fact that three structurally similar isoforms of TGFβ (TGFβ1, TGFβ2, and TGFβ3) were identified in humans[1,4,5]. While the three isoforms share amino acid homology, synthesis, receptors, and signal transduction mechanisms, individual knockout experiments demonstrate that their expression and proposed functions are distinct and non-redundant[2,6]. TGFβ1 is the most well-characterized isoform and is known to be abundant and ubiquitously expressed. However, it plays paradoxical roles in immune regulation depending on context. In the presence of IL-6, TGFβ1 can suppress Th1 and Th2 differentiation in favor of Th17 CD4$^+$ T cells, but in an anti-inflammatory environment, TGFβ1 can induce the formation of CD4$^+$CD25$^+$Foxp3$^+$ T regulatory cells (T$_{regs}$)[4,5,13–15]. TGFβ2 is primarily produced by neurons and glial cells in the nervous system and clinical investigations are underway with antisense oligonucleotides to target TGFβ2 in high-grade gliomas[4,5]. The tissue-specific expression patterns and functions of TGFβ3 are less well understood. TGFβ3 is thought to facilitate a scar-free fibrosis response, unlike TGFβ1 and TGFβ2, and depending on the context can mediate an anti-inflammatory response, with high levels of TGFβ3 correlating with reduced autoimmune encephalomyelitis in a mouse model[13], or proinflammatory state, as lung fibroblasts can produce a premetastatic niche in response to TGFβ3 expressed by breast cancer cells[4]. The context and cancer-specific roles of TGFβ isoforms, therefore, require further investigation. All three isoforms are secreted and sequestered as inactive homodimers in the extracellular matrix (ECM) and can be activated by a variety of mechanisms including integrins, reactive oxygen species, acid treatment, and enzymes that remodel the ECM[2,5,13,16]. The secretion and activation of TGFβ can be mediated by numerous cell types, including stromal components, immune cells, and tumor cells themselves, providing multiple therapeutic targets[5,17]. Therefore, the output of TGFβ signaling is highly contextual and varies across development, tissue, and tumor types[1,8].

Recent studies have focused on the effects of TGFβ inhibition in stroma-heavy cancers, such as CRC, urothelial carcinoma, and pancreatic ductal adenocarcinoma[7,12,18]. Both Tauriello et al. and Mariathasan et al. demonstrated that cancer-associated fibroblasts (CAFs), the most abundant stromal component, were the main source for all three isoforms of TGFβ in their models of CRC and urothelial carcinoma, respectively. Tauriello et al. suggest that TGFβ mediates treatment resistance in CRC by limiting T cell infiltration of tumors, resulting in immunologically cold tumors,

which can be turned hot or immune inflamed through TGFβ inhibition. However, the role of TGFβ signaling and its inhibition in stroma-poor cancers, such as melanoma, is yet to be explored. Furthermore, TGFβ can provide prognostic value as gene expression profiling of breast and urothelial cancer patients indicates that high TGFβ activity is associated with poor outcomes. In fact, high plasma levels of TGFβ1 correlate with reduced overall survival in CRC and breast cancer patients[4,5,12]. A defined TGFβ response gene signature is now being used to subtype CRC; consensus molecular subtype 4 (CMS4) CRC displays elevated TGFβ signaling activity, which confers a poor prognosis, higher relapse rates, and limited response to receptor tyrosine kinase therapy[4,7,8]. Currently, all available biologics and small-molecule inhibitors targeting the TGFβ pathway indiscriminately block all three isoforms and have been plagued by on-target toxicities, especially cardiac injury[2,19,20]. Additional studies are therefore needed to identify the TGFβ signature of different tumor types and to characterize biomarkers of response to therapy.

In this study, we aim to characterize the expression of TGFβ isoforms in the tumor microenvironment (TME) of fibroblast-poor tumors. Using mouse B16F10 melanoma (hereby referred to as B16 or B16 melanoma) and CT26 colon carcinoma as models, we demonstrate that TGFβ isoforms can be detected on tumor-infiltrating immune cells. We also show that isoform-specific inhibition of TGFβ is equivalent to pan-TGFβ inhibition in controlling B16 and CT26 tumor growth and, in combination with immune checkpoint blockade, can lead to improved tumor responses. Lastly, through T cell functional assays, we illustrate that isoform-specific blockade of TGFβ leads to activation of the adaptive immune system. These data suggest that inhibiting one isoform of TGFβ over another may lead to comparable therapeutic effects, while minimizing off-target side effects.

## Results

**The tumor microenvironment (TME) of B16 melanoma is not enriched with collagen or fibroblast cell types**. The TME, composed of cancer cells and supporting stromal cells, is now appreciated as a necessary component in carcinogenesis and is the target of many new anti-cancer strategies[21]. Cancer-associated fibroblasts (CAFs) are the most prominent stromal cell in many cancers, including breast, colorectal, and prostate, and are key determinants of tumor growth and invasion[22]. In this study, we use a transplantable model of mouse B16F10 melanoma to investigate the role of TGFβ in melanoma tumorigenesis. However, the microenvironment of this model of B16F10 mouse melanoma tumors is not considered to be dominated by fibroblasts. We compared the local milieu of B16F10 melanoma to 4T1 breast cancer as the immunosuppressive environment of 4T1 is thought to be due to the presence of CAFs[23].

To investigate whether there is a lack of CAFs in B16 melanoma, we conducted immunohistochemistry on B16 and 4T1 tumors 10 days post tumor implantation. Figure 1a shows an H&E stain of a similar section of 4T1 breast cancer (right) and B16 melanoma (left). We characterized the presence of fibroblasts in these sections using two different stains. Since fibroblasts are the main producers of collagen in the local environment we used picrosirius red as a way to quantify the amount of collagen fibers present in B16 versus 4T1 tumors[22]. Figure 1a shows tissue sections of B16 melanoma and 4T1 breast tumors stained with picrosirius red (PR). As quantified in Fig. 1b, the amount of PR-positive staining structures was significantly greater in the 4T1 breast tumors compared to the B16 melanoma tumors. This observation suggests that the environment of 4T1 breast tumors is more heavily defined by the presence of collagen, which is likely

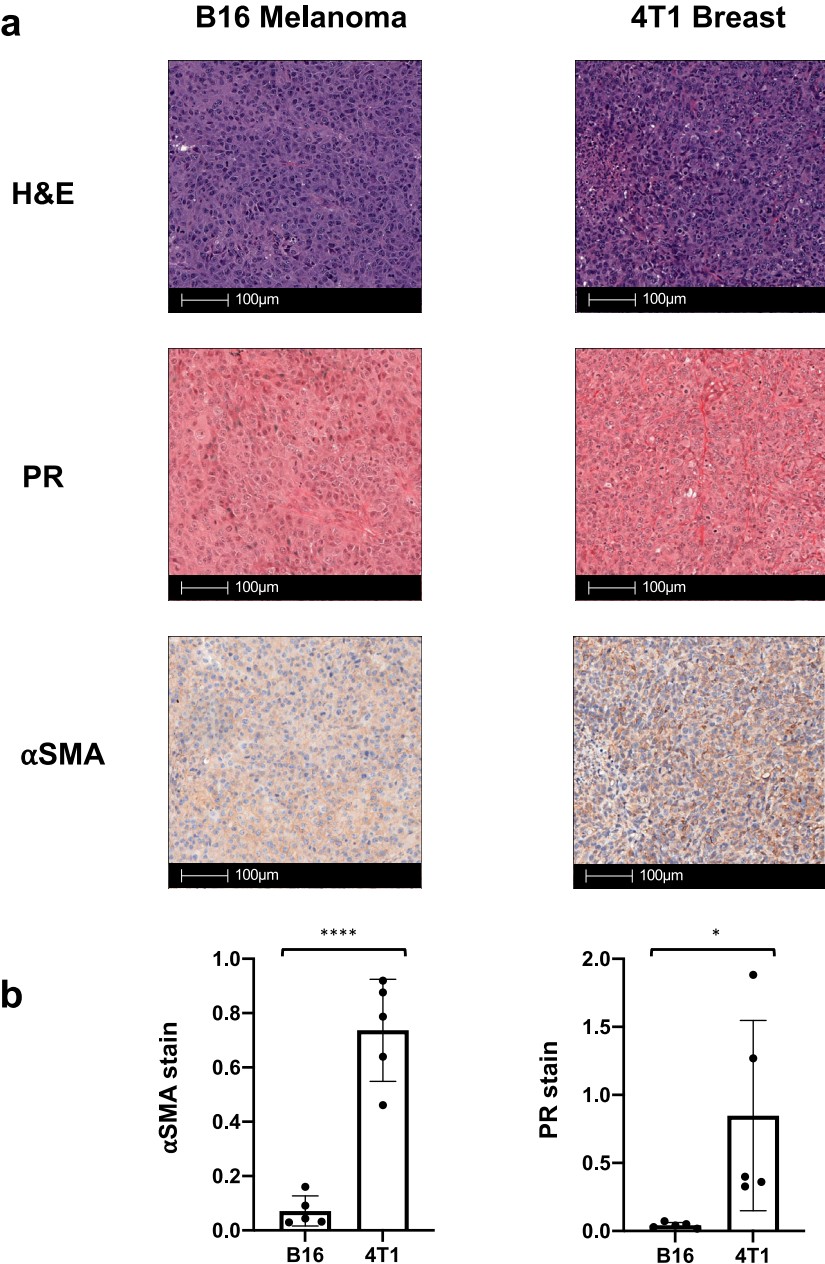

**Fig. 1 B16 melanoma is a stroma-poor tumor compared to 4T1 breast cancer.** Mice were implanted with 200,000 B16F10 cells (hereby referred to as B16 or B16 melanoma) injected intradermally or 200,000 4T1 cells injected subcutaneously ($n = 5$ mice/group). Tumors were harvested 10 days post tumor challenge and fixed for immunohistochemistry (IHC) prior to staining with picrosirius red (PR) and alpha-smooth muscle actin (αSMA). **a** Representative cross sections of B16 melanoma (left) and 4T1 breast (right) stained with hematoxylin and eosin (H&E, top), PR (middle), and αSMA (bottom). **b** Bar graphs demonstrate quantification of the staining of either PR or αSMA ± standard deviation (SD) following analysis by Halo software with supervision from a pathologist. *$p < 0.05$; ****$p < 0.001$.

primarily produced by surrounding fibroblasts. As a surrogate to quantify the number of fibroblasts, we used an alpha-smooth muscle actin (αSMA) stain IHC. αSMA is thought to be a marker of fibroblast activation and thus is expressed by many fibroblast subpopulations in the TME, including CAFs and myofibroblasts, as well as other cell types such as vascular cells and pericytes[22]. As shown in tissue sections in Fig. 1a and quantified in Fig. 1b, the density of αSMA immunoreactivity is greater in 4T1 breast tumors than in B16 melanoma tumors. While αSMA cannot be used to specifically identify CAFs in the tumor stroma, the greater proportion of αSMA positive immunoreactivity in 4T1 compared to B16 suggests that the tumor microenvironment of B16

melanoma contains a smaller proportion of fibroblast-like species[22].

We confirmed our findings in other stroma-poor and stroma-heavy murine tumor models using CT26 colon carcinoma and WG492 Braf[V600E] Pten[−/−] melanoma, respectively. Relative to WG492, CT26 demonstrated less positive immunoreactivity against αSMA and PR identifying CT26, similar to B16, as a stroma-poor tumor (Supplementary Fig. 1)[24]. Both B16 cells in-vitro and bulk B16 tumors produce all three isoforms of TGFβ as TGFβ isoform-specific mRNAs can be detected (Supplementary Fig. 2a). Taken together, our observations suggest that the primary cell type producing TGFβ in B16 melanoma and CT26

colon are not fibroblasts or its associated cell types. We, therefore, used B16 as a model to study the role of TGFβ in the context of non-desmoplastic tumors.

**TGFβ1 and TGFβ3 are highly expressed on myeloid cells in the tumor microenvironment.** As TGFβ plays a key role in regulating homeostatic pathways, it along with its receptors are thought to be expressed and secreted into the ECM by many cell types. In the TME, the main sources of TGFβ isoforms are the cancer cells, fibroblasts, and immune cells, including both lymphoid and myeloid cells[15,17]. TGFβ1 is the predominant isoform produced by the immune system and the expression of TGFβ2 and TGFβ3 is not well described[15]. Here, we investigate the expression of TGFβ isoforms and their role in B16 melanoma. To do so, we used two isoform-specific antibodies against TGFβ1 and TGFβ3 to analyze their protein expression by flow cytometry. These neutralizing antibodies are highly specific to the active form of either TGFβ1 or TGFβ3 as shown in supplementary Fig. 3a and can be used for blocking studies. In addition, because these antibodies only detect the active form of TFGβ, the expression pattern is similar even when the staining is done via surface or intracellular staining (Supplementary Fig. 3b). There are no suitable commercially available mouse TGFβ2-specific monoclonal antibodies; moreover, TGFβ2 expression is thought to be mostly limited to the nervous system[4,5].

Most of the published studies on TGFβ focus on its role in stroma-heavy tumors where CAFs are cited as the main source for all TGFβ isoforms[7,12]. However, as shown in Fig. 1 and Supplementary Fig. 1, the stroma of mouse B16 melanoma and CT26 colon carcinoma is not defined by the presence of fibroblasts. While CAFs, thought to be the main producers of TGFβ, are not the primary component of the stroma in mouse B16 melanoma, the mRNA expression of all three TGFβ isoforms was detected in B16 cells in vitro and in vivo. TGFβ3 mRNA was the predominant isoform detected both among B16 cells in-vitro and from bulk tumors (Supplementary Fig. 2a). Since the CD45-population from isolated B16 tumors demonstrated lower protein expression of TGFβ1 and TGFβ3 compared to tumor-infiltrating Ly6C⁺ high monocytes (Supplementary Fig. 2b, c), we focused on characterizing the isoform-specific expression of TGFβ on tumor-infiltrating lymphoid and myeloid immune cells.

Flow cytometry analysis was conducted on immune cells from B16 tumors harvested 11 days after tumor implantation. The immune infiltrate at this time point is dominated by both CD8⁺ T cells and CD11b⁺CD11c⁺ myeloid/dendritic cells (DCs) (Fig. 2a). Based on mean fluorescence intensity (MFI), TGFβ1 and TGFβ3 are highly expressed by myeloid lineage cells, specifically Ly6C⁺ high monocytes and CD11b⁺CD11c⁺DCs (Fig. 2b, c). A similar trend is found peripherally in the spleens of mice at the same time point with CD8⁺CD11c⁺DCs and Ly6C⁺ high monocytes expressing the highest levels of TGFβ1 and TGFβ3 isoforms (Supplementary Fig. 4b, c). Further along in tumor progression, the expression of TGFβ isoforms continues to be greatest on myeloid cells. At day 15 post tumor implantation, TGFβ1 and TGFβ3 are highly expressed on CD8⁺DCs and Ly6G⁺ granulocytes (Supplementary Fig. 5b, c). While the phenotypic markers for granulocytes (CD11b⁺Ly6G⁺) and monocytes (CD11b⁺Ly6Cʰⁱᵍʰ) are the same as their myeloid-derived suppressor cell (MDSC) counterparts, these cells are not immunosuppressive in B16 melanoma when compared to other tumor models enriched with MDSCs[25,26].

Compared to B16 melanoma, CT26 colon cancer also demonstrated higher expression of TGFβ isoforms on infiltrating myeloid cells compared to lymphoid cells. However, in CT26 colon cancer, there is a relatively greater expression of TGFβ1

than TGFβ3 by comparison of MFI values (Supplementary Fig. 6). While the microenvironment of B16 has relatively equal expression of both TGFβ isoforms, CT26 is dominated by TGFβ1 expression and illustrates a distinct TGFβ signature.

To further verify the specificity of these isoform-specific antibodies and correlate the expression of TGFβ mRNA with its protein production in particular immune cells, we conducted standard flow cytometry and RNA primeflow to co-stain for TGFβ isoforms at both the protein and mRNA level. Using fluorophore-conjugated complementary mRNA probes along with fluorophore-conjugated antibodies against TGFβ1 and TGFβ3 proteins, we are able to co-stain for these TGFβ isoforms on specific immune cells (Supplementary Fig. 7a, b). Overall, our data demonstrate that there is differential expression of TGFβ isoforms on immune cell populations in the TME of B16 melanoma.

**Isoform-specific TGFβ inhibition can control B16F10 melanoma and CT26 colon tumor growth.** Since we found that both TGFβ1 and TGFβ3 isoform expression were detectable at 11 days post tumor implantation, a time point at which the B16F10 tumors are palpable and well established, we began treatment with isoform-specific anti-TGFβ therapy at this time. We confirmed in vivo inhibition of canonical TGFβ signaling via the reduction in phosphorylated SMAD2/3 expressed in tumor-infiltrating CD45⁺ immune cells following isoform-specific and pan-TGFβ inhibition (Supplementary Fig. 8). Using a previously published protocol for anti-TGFβ therapy[16], the antibodies were delivered via intraperitoneal injection (200 μg/mice) beginning 11 days after tumor implantation and continuing every other day for a total of eight doses (Fig. 3a). Compared to untreated control animals, both isoform-specific TGFβ blockade and pan-TGFβ inhibition (with 1D11) resulted in delayed B16 tumor growth. Anti-TGFβ3 therapy resulted in the greatest delay in tumor growth (62.3% reduction in tumor size compared to control), followed by anti-TGFβ1 therapy (49.68%) and pan-TGFβ blockade (37.44%) calculated 24 days post tumor implantation (Fig. 3b, c). However, none of these monotherapies resulted in improved overall survival.

We investigated the anti-tumor efficacy of isoform-specific TGFβ inhibition in CT26 colon cancer, another stroma-poor tumor model (Supplementary Fig. 1). Analysis of the immune infiltrate in CT26 demonstrated higher expression of TGFβ1 compared to TGFβ3 based on MFI values (Supplementary Fig. 6). In CT26 isoform-specific inhibition with TGFβ1 and pan-TGFβ inhibition were effective at delaying tumor growth while TGFβ3 inhibition had no anti-tumor effect (Fig. 3d). Our data demonstrate that isoform-specific inhibition is effective at delaying tumor growth in stroma-poor tumors such as B16 and CT26. In addition, these data support the idea that each tumor type may have a different dominant TGFβ isoform that hinders anti-tumor immunity.

**Isoform-specific TGFβ inhibition leads to CD8⁺ T cell activation.** The anti-tumor effects observed in vivo in B16 melanoma led us to hypothesize that isoform-specific TGFβ inhibition can induce an anti-tumor response that is in part immune-dependent. We harvested tumors from control animals and animals treated with either anti-TGFβ1, anti-TGFβ3, and 1D11 post 4 doses of therapy (Fig. 4a). At this time point, there was an increase in CD45⁺ total immune cells and CD8⁺ T cell infiltration in B16 tumors in treated animals compared to untreated controls. This intra-tumoral augmentation of CD45⁺ immune infiltration and CD8⁺ T cells was significant in animals treated with either anti-TGFβ3 or 1D11 (Fig. 4b). However, no significant differences

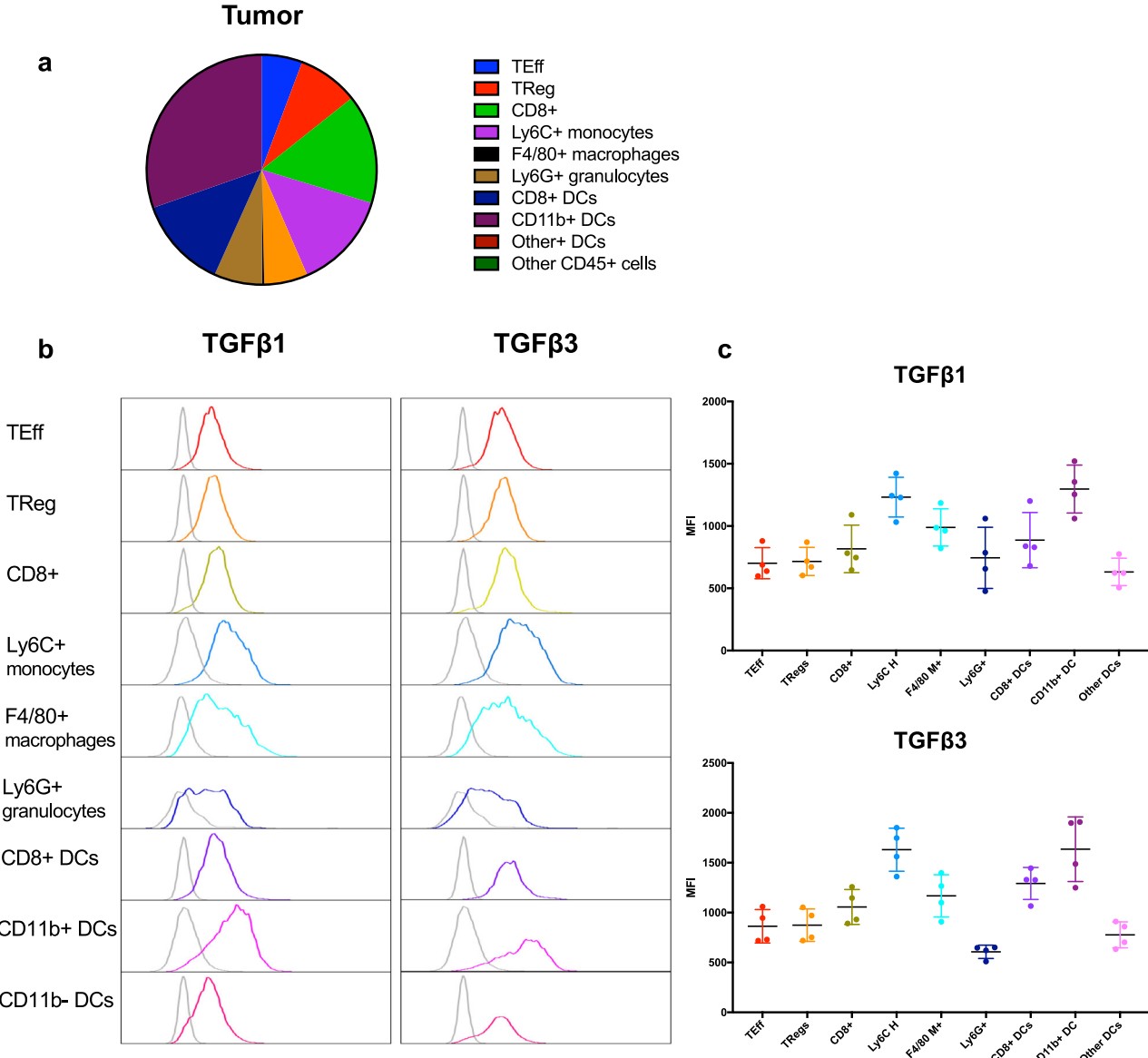

**Fig. 2 Tumor-infiltrating myeloid immune cells and dendritic cells (DCs) display the highest levels of TGFβ isoform expression.** B16 tumors from mice were harvested 11 days following tumor implantation. After creating single-cell suspensions, tumors underwent processing and staining for flow cytometry analysis as described in "Methods". **a** Breakdown of the immune infiltrate in B16 tumors of mice harvested 11 days after tumor implantation. **b** Representative histograms displaying mean fluorescence intensity (MFI) of either TGFβ1 (left panel) or TGFβ3 (right panel) on specific tumor-infiltrating immune cells as detected by flow cytometry. The light gray peak represents each cell type's fluorescence minus one (FMO) and was used to determine positive expression, indicated by the colored peak. **c** Representative graph illustrating the relative expression of TGFβ1 (top graph) or TGFβ3 (bottom graph) by various lymphocytic and myeloid cell types in the tumor microenvironment 11 days after tumor implantation. Data (*n* = 5 mice/group) are displayed as MFI ± SD. Data are representative of three independent experiments. MFI is measured in arbitrary units and is a variable used to measure relative expression levels of staining antibodies, in this case of TGFβ1 and TGFβ3 protein expression, on tumor-infiltrating immune cells. FMO controls were derived by staining the immune cells with all the fluorophores minus one fluorophore, in this case, the fluorophore (Alexa Fluor 647) that was conjugated to TGFβ1 and TGFβ3. The pattern of intracellular TGFβ1 and TGFβ3 expression is similar to the surface staining pattern demonstrated in Supplementary Fig. 3b.

were detected in the quantities or activation status of CD4+ T effector cells (Foxp3−) or CD4+ Tregs (Foxp3+). Following four doses of anti-TGFβ therapy, we observed little change in the myeloid compartment (Supplementary Fig. 9) except for a significant decrease in suppressive macrophages in mice treated with anti-TGFβ3 or 1D11 (Supplementary Fig. 10); therefore we focused on further characterizing the T cell response.

As CD8+ T cells appeared to dominate the immune infiltrate and are thought to play a major role in mediating the anti-tumor response to B16 melanoma, we characterized the activation status

of CD8+ tumor-infiltrating lymphocytes (TILs) following anti-TGFβ treatment[25]. Using Granzyme B expression as a surrogate marker of CD8+ cytolytic function, we found that CD8+ T cells in tumors that had received either isoform-specific or pan-TGFβ inhibition expressed higher levels of Granzyme B compared to untreated tumors (Fig. 4c, d). This finding of increased Granzyme B in CD8+ T cells following TGFβ inhibition is consistent with results previously published in models of metastatic CRC and urothelial cancer[7,12]. The enhanced cytolytic ability of CD8+ T cells could account for the delay in tumor growth seen with

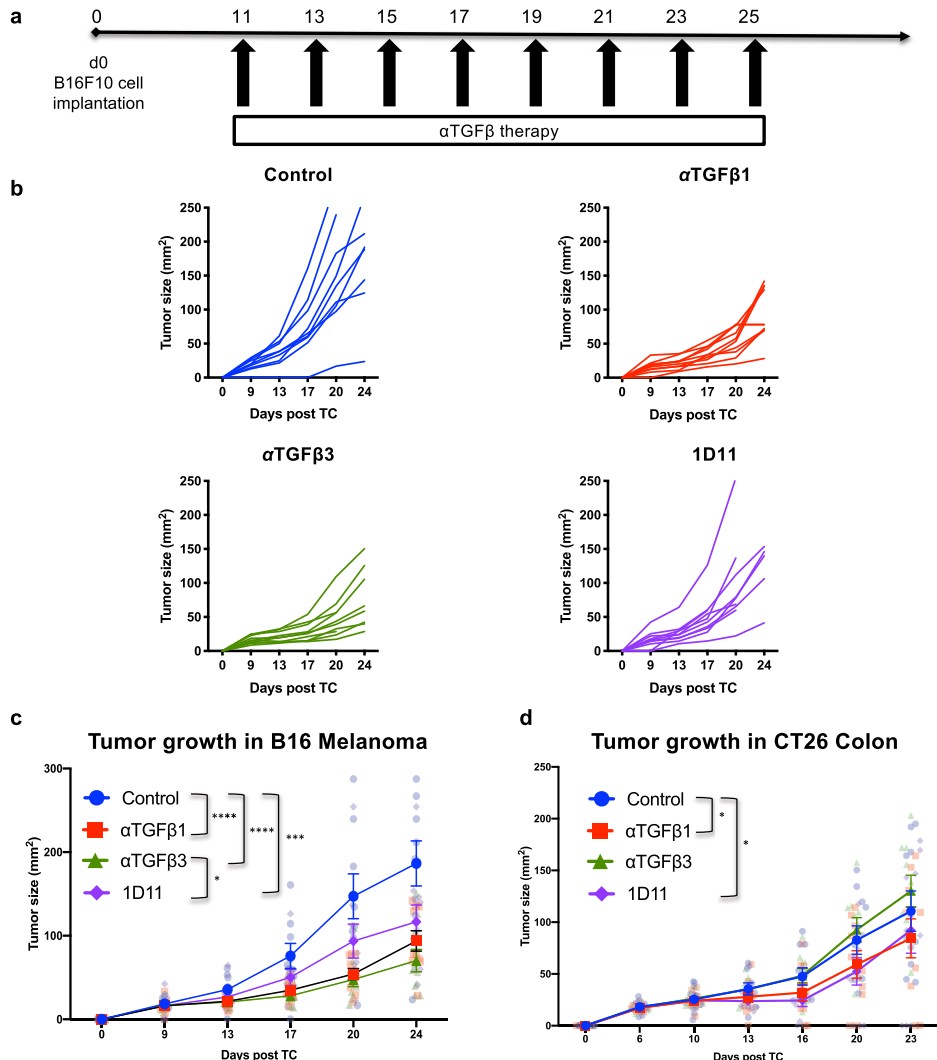

**Fig. 3 Isoform-specific TGFβ inhibition is effective at delaying B16 tumor growth. a** Therapy regimen beginning 11 days post tumor implantation with 250,000 B16 cells. Mice were treated with 8 doses of anti-TGFβ therapy given every other day via intraperitoneal injection at 200 μg/mouse ($n = 10$ mice/group). **b** Individual tumor growth curves for control and treated groups in B16 melanoma. Data are representative of three independent experiments. **c** Tumor size (measured as surface area in mm²) of untreated animals and animals treated with anti-TGFβ1, anti-TGFβ3, and 1D11 (pan-TGFβ inhibition) in B16 melanoma (left) (**d**) and CT26 colon tumors (right). Data is displayed as ± standard error the mean (SEM). Statistics were calculated using 2-way ANOVA 24 days post tumor implantation for B16 and 23 days post tumor implantation for CT26. Only statistically significant differences among untreated and treated groups are shown. *$p < 0.05$; ***$p < 0.0005$; ****$p < 0.0001$.

TGFβ inhibition compared to untreated animals. Furthermore, the TGFβ blockade affected the expression of additional CD8+ T cell activation markers. CD8+ TILs from animals that had received either anti-TGFβ3 or 1D11 treatment demonstrated significant increases in the proliferation marker Ki67+ (Fig. 4d). While these treatment groups did show increased expression of PD-1+, a marker of T cell exhaustion, they also displayed higher percentages of both PD-1+GrzB+ T cells, suggesting that these cells are antigen-experienced and have greater cytolytic capabilities (Fig. 4d). More importantly, ex vivo analysis of CD8+ T cells revealed that isoform-specific TGFβ inhibition is able to enhance the cytotoxic and antigen-specific responses of these immune cells. We performed a killing assay with CD8+ T cells, purified from the spleens of control, and treated animals. These cells were then co-cultured with B16 cells at an effector: target ratio of 50:1 for 48 h and demonstrated enhanced killing in all conditions treated with TGFβ blockade. The greatest level of B16 melanoma direct killing was seen in CD8+ T cells from animals treated with anti-TGFβ3 (47.8% killing), followed by those treated with

anti-TGFβ1 (34.1% killing) and 1D11 (21.8% killing) compared to 10.6% killing in untreated controls (Fig. 4e). This mirrors the anti-tumor efficacy observed in Fig. 3b, c. Isoform-specific TGFβ inhibition also improved melanoma-specific cytokine responses of CD8+ T cells. TGFβ1 and TGFβ3 blockade resulted in increased interferon-γ production by purified CD8+ T cells when co-cultured ex vivo with irradiated B16 cells; however, pan-TGFβ inhibition with 1D11 was unable to elicit antigen-specific cytokine responses from isolated CD8+ T cells (Fig. 4f).

**Abrogation of anti-tumor effect following CD8+ T cell depletion.** Given the enhancement in CD8+ T cell effector phenotype and function, we depleted CD8+ T cells in treated animals to determine whether they were critical for mediating the anti-tumor effect of TGFβ inhibition. Depletion of CD8+ T cells was confirmed in both the tumor and spleens of treated animals following two doses of anti-CD8 antibody (Fig. 5a). Compared to untreated control animals, depletion of CD8+ T cells resulted in

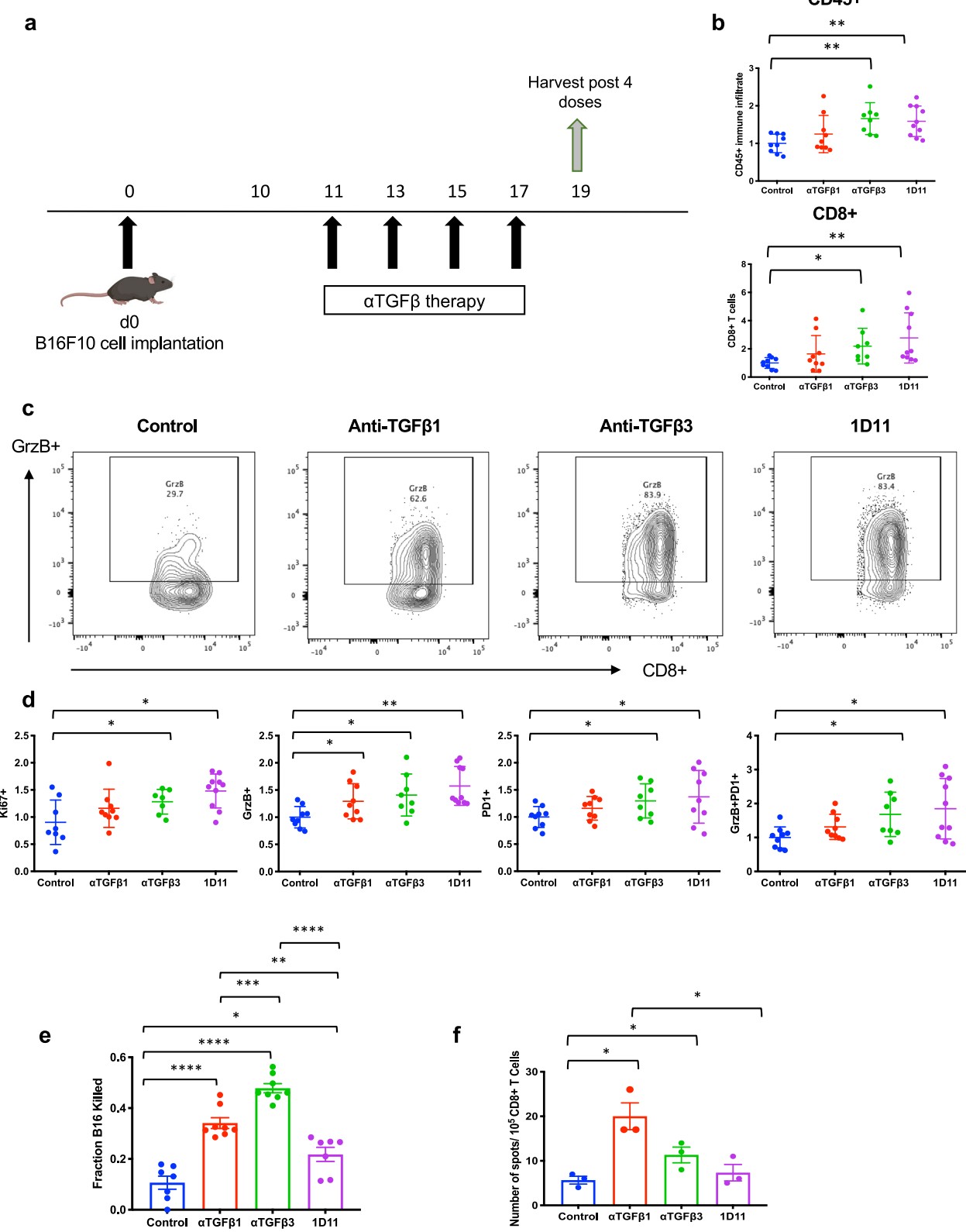

enhanced B16 melanoma tumor growth (Fig. 5b). With either isoform-specific TGFβ or pan-TGFβ inhibition, depletion of CD8+ T cells abrogated the tumor protective effect conferred by anti-TGFβ therapy (Fig. 5b). This and the data are shown in Fig. 4 suggest that CD8+ T cells are critical in mediating the anti-tumor effects in B16 melanoma by TGFβ inhibition.

**Isoform-specific TGFβ inhibition together with ICB improves B16 tumor control.** Given the lack of curative responses with anti-TGFβ therapy alone against B16 melanoma, we hypothesized that the established tumors harbored an immunosuppressive microenvironment that limited CD8+ anti-tumor activity. Recent data analyzing patients with metastatic urothelial cancer links

**Fig. 4 Isoform-specific TGFβ inhibition induces CD8$^+$ T cell activation. a** Experimental schema used to analyze B16 tumor immune infiltrates. Tumors of control and treated animals ($n = 5$ mice/group) were harvested following four doses of anti-TGFβ therapy and underwent flow cytometric processing and analysis as described in "Methods". **b** Plots showing changes in CD45$^+$ tumor-infiltrating immune cells (top panel) and CD8$^+$ tumor-infiltrating lymphocytes (TILs) (bottom panel) post four doses of anti-TGFβ treatment. Data represent pooled values from two independent experiments ($n = 5$ mice/group) that were normalized to the control and are displayed as fold change compared to the control ± SD initially gated on CD45$^+$ immune cells and subgated on CD8$^+$ T cells. **c** Representative flow cytometry plots showing Granzyme B$^+$ (GrzB) expression in CD8$^+$ TILs, which were gated from CD45$^+$ live cells. **d** Activation of CD8$^+$ TILs from control and anti-TGFβ treated groups showing changes in Ki67$^+$, GrzB$^+$, PD-1$^+$ and GrzB$^+$PD-1$^+$CD8$^+$ T cells post 4 doses of therapy. Data represent pooled values from two independent experiments normalized to the control ($n = 5$ mice/group) and is displayed as fold change compared to the control ± SD gated on CD8$^+$ T cells. **e** CD8$^+$ T cells were purified from the spleens of control and treated animals according to the experimental setup shown in a. For the killing assay, CD8$^+$ T cells were plated with B16 cells at a ratio of 50:1 effector: target for 48 h. The remaining B16 cells were measured using a clonogenic assay 1 week later. Data are representative of pooled values from two independent experiments ($n = 5$ mice/group) normalized to the highest count. B16 killing was normalized to the highest count and the fraction of killing is displayed as ± SEM. **f** For the IFN-γ EliSpot, CD8$^+$ T cells were plated with irradiated B16 cells at a ratio of 2:1 for 24 h. IFN-γ production was quantified using the ImmunoSpot platform. Data are representative of two independent experiments ($n = 3$ mice/group) and is plotted as ±SEM. For all panels only statistically significant differences among untreated and treated groups is shown. *$p < 0.05$; **$p < 0.001$; ***$p < 0.0005$; ****$p < 0.0001$.

PD-L1 expression with TGFβ signaling. PD-L1 expression on immune cells was associated with response to anti-PD-L1 therapy, whereas non-responders showed high expression of TGFβ pathway genes[12]. In our model of established B16 tumors, PD-L1 expression significantly increased on tumor-infiltrating CD11b$^+$ and CD11c$^+$ cells as a result of TGFβ1 and TGFβ3 isoform-specific and pan-TGFβ inhibition (Supplementary Fig. 11a, b). Specifically, Ly6C$^+$ high monocytes and CD11c$^+$CD8$^+$DCs, the cell types that were found to have high expression of both TGFβ1 and TGFβ3, demonstrated an increase in PD-L1 expression following isoform-specific or pan-TGFβ blockade (Supplementary Fig. 11c). Along with increased ligand expression, we found that the receptor (PD-1) is also upregulated on CD8$^+$ T cells (Fig. 4d). Therefore, activation of the PD-L1–PD-1 axis may create an immunosuppressive environment that counteracts the cytotoxic effect of T cells within the tumor, preventing complete regression.

Immune checkpoint blockade (ICB) with anti-PD-1/anti-PD-L1 and anti-CTLA-4 is now a standard option for the clinical management of several cancers, especially metastatic melanoma. However, not all patients have robust and durable responses to these therapies[27]. In order to target non-redundant immune regulatory pathways, such as those activated by TGFβ and PD-1/PD-L1 or CTLA-4 expression, we proposed combining ICB with isoform-specific TGFβ inhibition in order to improve CD8$^+$ T cell responses and anti-tumor immunity. Animals were treated with isoform-specific TGFβ inhibition and ICB according to the schedules outlined in Fig. 6a. In all cases of TGFβ inhibition, anti-CTLA-4 therapy was superior at controlling tumor growth. The addition of isoform-specific or pan-TGFβ inhibition did not significantly improve tumor control (Fig. 6b). Similarly, anti-CTLA-4 treatment resulted in a greater overall survival compared to anti-TGFβ therapy alone; however, the combination of the two therapies with isoform-specific inhibition did impact overall survival over that which was achieved with anti-TGFβ monotherapy (Supplementary Fig. 12).

Isoform-specific TGFβ inhibition and pan-TGFβ blockade both showed efficacy when combined with anti-PD-1. While anti-PD-1 treatment alone effectively controlled B16 tumor growth in comparison to TGFβ monotherapy; the combination of anti-TGFβ1 with anti-PD-1 therapy was superior to either treatment alone in delaying tumor growth (Fig. 6c). The combination therapy also improved overall survival compared to anti-TGFβ1 treatment alone (Supplementary Fig. 12). In comparison to anti-TGFβ3 or 1D11 treatment, anti-PD-1 alone was able to produce a significant anti-tumor effect. However, the combination of either therapy with anti-PD-1 resulted in a greater delay in B16 tumor growth compared to anti-TGFβ monotherapy (Fig. 6c). Only the combination of either isoform-specific TGFβ or pan-TGFβ

inhibition with PD-1 blockade produced an improvement in overall survival compared to TGFβ inhibition alone (Supplementary Fig. 12). It is interesting to note that in this model of B16 melanoma, isoform-specific inhibition with anti-TGFβ1 therapy and anti-PD-1 is superior to anti-PD-1 alone; however, TGFβ inhibition in combination with anti-CTLA-4 therapy did not produce synergistic effects.

## Discussion

In this study, we characterized the immune cell expression of TGFβ1 and TGFβ3 in non-stromal rich tumors using mouse melanoma and colon carcinoma as model systems. Our results indicate that in stroma-poor tumors infiltrating myeloid cells are the main producers of TGFβ. In CT26 the predominant isoform expressed by infiltrating immune cells is TGFβ1 (Supplementary Fig. 6) and correspondingly anti-TGFβ1 demonstrated superior tumor control (Fig. 3d). Without ample ligand to inhibit in CT26, TGFβ3 inhibition is unable to suppress tumor growth. In B16 there is an equal expression of both isoforms on infiltrating immune cells (Fig. 2c and Supplementary Fig. 6) and isoform-specific inhibition of either TGFβ1 or TGFβ3 curbed tumor growth (Fig. 3b, c). These results illustrate that each stroma-poor tumor type has a specific TGFβ signature with different balances of TGFβ1 versus TGFβ3 in the local microenvironment. Canè et al. showed similar results with TGFβ1 inhibition potentiating the anti-tumor effect of prophylactic vaccination with irradiated CT26 cells[28] and Terabe et al. demonstrated that inhibition of TGFβ1 and TGFβ2 can reduce tumor burden in lungs with a metastatic CT26 model[29]. Our results illustrate the high expression of TGFβ1 in CT26 tumors coupled with in vivo efficacy data and previously published studies establish CT26 as having a TGFβ1 signature responsive to TGFβ1 inhibition. Canè et al. also demonstrated that TiRP melanoma is characterized by high expression of both TGFβ1 and TGFβ3 transcripts which they found are primarily produced by the tumor cells and stroma (defined as non-tumor cells), respectively[28]. Using RNA-sequencing data from The Cancer Genome Atlas, Martin et al. found that while TGFβ1 mRNA is the most prevalent isoform expressed in the majority of human cancers, certain cancer types, such as breast, mesothelioma, and prostate, are defined by high expression of both TGFβ1 and TGFβ3 mRNA[30]. Further studies are needed to characterize the TGFβ signature of different tumor types and how it impacts the efficacy of anti-tumor therapies.

TGFβ has garnered interest recently as a target against cancer. While normal melanocytes are responsive to the cytostatic effects of TGFβ, melanoma cells are able to escape the anti-proliferative effects of TGFβ through a poorly understood mechanism. Alterations in TGFβ pathway receptors or signal transducers,

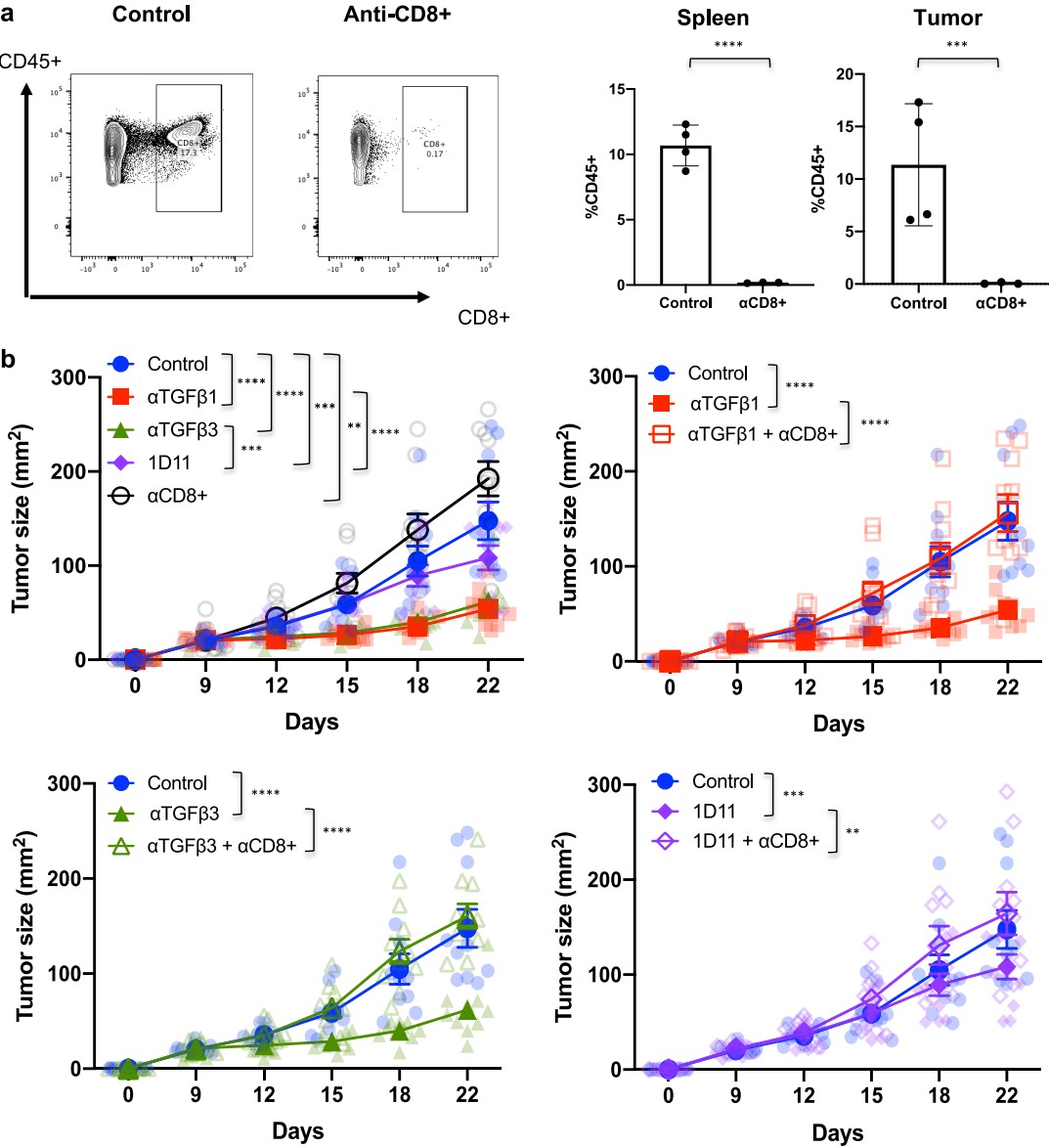

**Fig. 5 Depletion of CD8+ T cells abrogates the anti-tumor effect of TGFβ inhibition.** Mice were tumor challenged with B16 cells and treated according to the therapy regimen outlined in Fig. 3a ($n = 10$ mice/group). **a** Following two doses of anti-CD8+ therapy, mice were sacrificed and spleens and tumors were harvested for flow cytometry analysis. **a** Representative flow cytometry plots (left) demonstrating CD8+ T cells in the tumors of control and anti-CD8+-treated animals. Plots (right) showing CD8+ T cell depletion in the spleens and tumors of control and anti-CD8+-treated animals. Data plotted as mean ± SD. **b** Mice received anti-CD8+ therapy given via intraperitoneal injection twice weekly for the duration of anti-TGFβ therapy. Tumor growth (mm2) and overall survival were monitored. Tumor growth curves are shown for isoform-specific anti-TGFβ and pan-anti-TGFβ therapy in combination with anti-CD8+ treatment ± SEM. Statistics were calculated at 22 days post tumor implant by 2-way ANOVA. Only statistically significant differences between groups are displayed. Data are representative of two independent experiments. **p < 0.005; ***p < 0.0005; ****p < 0.0001.

such as TGFβ receptors or SMAD proteins, are rarely detected and do not account for melanoma's resistance to TGFβ-mediated growth suppression[31]. The expression of TGFβ is not restricted to tumor cells and fibroblasts, as many immune and non-immune cells are capable of producing and activating TGFβ. While TGFβ is known to directly suppress T cells as T cell-specific deletions of TGF-βRII or TGF-βRI result in rapid lethal inflammatory disease, myeloid cells are thought to be a major source of TGFβ in the TME[32]. In a study of 4T1 mouse breast cancer, the authors found that Gr1+CD11b+ cells are a major source of TGFβ in the TME, and depletion of these cells abrogates the anti-tumor effect of pan-TGFβ inhibition[33]. Similarly, another study determined that

Gr1+CD11b+ cells and to a lesser extent CD11c+ cells isolated from the spleens of mice injected with a fibrosarcoma cell line were the major producers of TGFβ1 ex vivo[34]. In agreement with these findings, we found that TGFβ1 and TGFβ3 are highly expressed on tumor-infiltrating myeloid-dendritic cells early in tumorigenesis in B16 (Fig. 2). Non-lymphoid cells isolated from the spleens of tumor-bearing mice also demonstrate high expression of these TGFβ isoforms at the same time point (Supplementary Fig. 4). Together with our findings, it appears that non-lymphoid cells, in particular myeloid cells, are critical sources of TGFβ1 and TGFβ3 across different tumor types. The similar staining pattern of TGFβ isoforms between surface

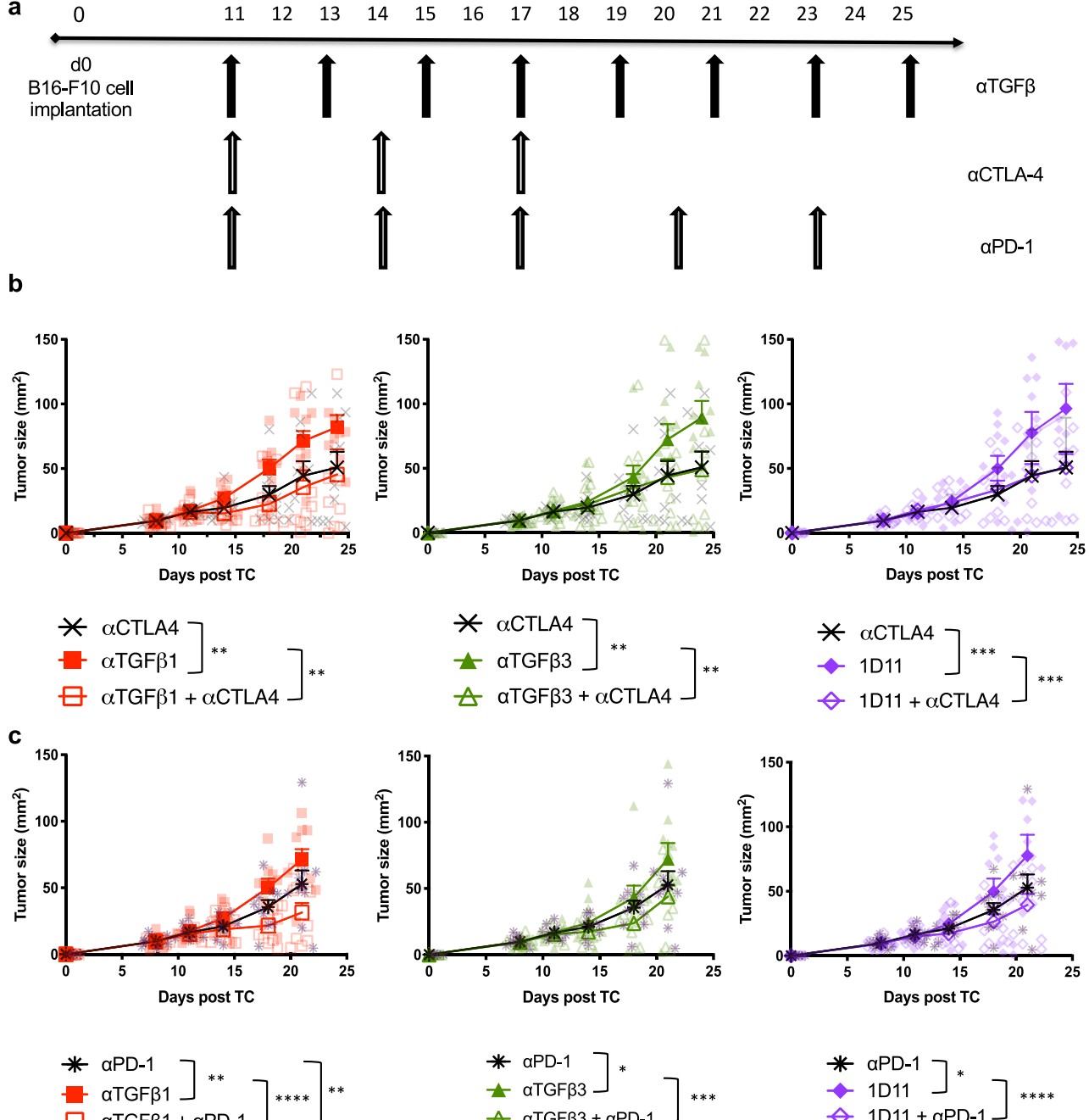

**Fig. 6 TGFβ1 inhibition in combination with immune checkpoint blockade delays B16 tumor growth. a** Treatment regimen illustrating the schedule of delivery of anti-TGFβ (pan or isoform-specific inhibition) with either anti-CTLA-4 or anti-PD-1 therapy beginning 11 days post tumor implantation. Anti-TGFβ therapy was given via intraperitoneal injection (200 μg/mouse) every other day for a total of 8 doses. Anti-CTLA-4 (clone 9H10) was given via intraperitoneal injection (100 μg/mouse) every 3 days for a total of 3 doses. Anti-PD-1 was given via intraperitoneal injection (250 μg/mouse) every 3 days for a total of 5 doses. **b** Tumor growth curves for indicated treatment combinations with anti-CTLA-4. **c** Tumor growth curves for indicated treatment combinations with anti-PD-1. Data shown are representative of two independent experiments ± SEM. Statistics were calculated 24 days post tumor implant (n = 10 mice/group) using 2-way ANOVA. Only statistically significant differences among untreated and treated groups are shown. *$p < 0.05$; **$p < 0.005$; ***$p < 0.0005$; ****$p < 0.0001$.

(Supplementary Fig. 3b) and intracellular staining (Fig. 2) demonstrates that most of the active form of TGFβ is found on the surface of tumor-infiltrating immune cells.

We have demonstrated that B16F10 melanoma tumors are highly infiltrated with activated CD8[+] T cells, which represent critical targets for TGFβ inhibition in the TME[25]. TGFβ is known to have potent inhibitory effects on T cell proliferation, differentiation and effector function. TGFβ is necessary to mediate

immune tolerance via T cells as T cell-specific deletions of *Tgfbr2* phenocopy the systemic inflammatory disorder that results from *Tgfb1* knockout mice[35–37]. Recent data indicate that TGFβ, either through acting as a surface-bound ligand on Tregs or via increasing the CD4[+]Treg/CD4[+]Th ratio, impairs the anti-tumor response in melanoma and other skin cancers via regulatory T cells[14,38]. Our results using isoform-specific TGFβ inhibition recapitulate many existing studies using pan-TGFβ inhibition

both in chronic viral and transplantable tumor mouse models. We showed that isoform-specific inhibition of TGFβ and pan-TGFβ inhibition increase $CD8^+$ T cell infiltration into B16 melanoma tumors and enhance their effector phenotype. Both anti-TGFβ1 and anti-TGFβ3 result in increased Granzyme B expression compared to untreated animals, which results in enhanced cytolytic activity demonstrated through an ex vivo killing assay. Furthermore, isoform-specific inhibition as opposed to pan-TGFβ inhibition enhanced antigen-specific T cell cytokine responses (Fig. 4). These results are similar to those seen in a chronic viral model of LCMV with mice harboring a dominant-negative form of TGFβ receptor II in T cells. The authors demonstrate that attenuated TGFβ signaling resulted in the accumulation and persistence of virus-specific $CD8^+$ T cells that acquired enhanced effector functions such as secretion of interferon-γ, TNF-α, and IL-2 following LCMV peptide stimulation[39].

Similarly, recent studies in a genetically engineered mouse model of colorectal carcinoma demonstrated increased expression of PD-1 and Granzyme B on tumor-infiltrating $CD8^+$ T cells following galunisertib (a small-molecule selective inhibitor of TGF-β receptor type I) administration, which was further enhanced in combination with anti-PD-L1[7]. In the spontaneous TRAMP model of prostate cancer, expression of a dominant-negative form of TGFβ receptor II in T cells exhibited tumor protection that was associated with enhanced $CD8^+$ T cell infiltration and Granzyme B expression[35,40]. These findings along with our data showing an abrogation of tumor protection following $CD8^+$ T cell depletion (Fig. 5), suggest that isoform-specific and pan-TGFβ inhibition mediate anti-tumor efficacy via $CD8^+$ T cells and that targeting TGFβ1 or TGFβ3 can result in enhanced $CD8^+$ T cell effector functions. While TGFβ1 is thought to be the primary isoform expressed in the immune system, our data suggest that TGFβ1 and TGFβ3 are both highly detectable in the tumor microenvironment and may play alternate roles in regulating T cell biology, thus offering alternate targets for anti-cancer therapy.

The advent of immunologic checkpoint blockade has altered the oncology landscape. While patients have experienced unprecedented responses to immune-modulating agents such as anti-CTLA-4 and anti-PD-1 therapy, only 20–60% of patients have durable clinical responses to immune checkpoint blockade[41]. Further investigation is required to understand mechanisms of resistance that prevent durable clinical responses in many patients. Through its effects on tumor cells and the surrounding TME, TGFβ is thought to be a critical catalyst of such immune tolerance[42]. Recent studies in stroma-heavy tumors such as microsatellite-stable (MSS) CRC and urothelial cancer demonstrate the ability of TGFβ inhibition to turn these immunologically cold tumors, with poor $CD8^+$ T cell infiltration, into hot tumors with an increased infiltration of cytotoxic $CD8^+$ T cells. In these models, fibroblasts are proposed as the main producers of TGFβ, which physically limit T cell infiltration into tumors[7,12]. As demonstrated in Fig. 1, B16 melanoma is a stroma-poor tumor with a low density of αSMA staining and is known to be highly infiltrated with T cells[25]. A similar case is found in human melanoma samples with 2–4% of melanomas considered as desmoplastic[43,44]. Rather than the stroma serving as a physical barrier to T cells in the context of melanoma, we propose that tumor-infiltrating myeloid cells are the main producers of TGFβ, which suppress the activation and cytotoxic function of local $CD8^+$ T cells. However, the producers of TGFβ are not necessarily the cells activating it and additional research is required to elucidate which immune and non-immune cells are involved in the TGFβ signaling pathway in the TME.

The cytotoxic activity of tumor-infiltrating $CD8^+$ T cells may be diminished by the upregulation of PD-1 and PD-L1, as seen on infiltrating immune cells (Fig. 4d and Supplementary Fig. 11), as well as the downstream effects of TGFβ signaling. As anti-CTLA-4 and anti-PD-1/PD-L1 are established therapies that are utilized in the clinic, we hypothesized that the addition of ICB to anti-TGFβ therapy has the potential to induce durable complete responses that are infrequently seen with TGFβ inhibition alone in transplantable cancer models[12,16]. Tauriello et al. showed in a genetically engineered model of MSS CRC that $PD-1^+$ and $PD-L1^+$ expression increased on $CD45^+$ infiltrating immune cells following galunisertib administration. The addition of anti-PD-L1 therapy to galunisertib prolonged overall survival to greater than a year post treatment[7]. A corresponding study by Mariathasan et al. demonstrated that the combination of pan-anti-TGFβ and anti-PD-L1 therapies induced complete regression (70%) in a transplantable model of mouse mammary carcinoma compared to 0% and 10% with either therapy alone, respectively[12]. Similarly, ICB-resistant prostate cancer that is metastatic to bone was shown to regress in combination with TGFβ inhibition due to increased Th1 $CD4^+$ and $CD8^+$ T cells[45].

The development of bifunctional antibody ligand traps comprised of an antibody targeting an immune checkpoint at one end and entrapping soluble TGFβ on the other suggests that one mechanism to overcome local immune tolerance at the TME is through combination therapies[42]. Our data demonstrate that PD-1 blockade together with either pan or isoform-specific TGFβ inhibition enhances overall survival compared to anti-TGFβ monotherapy (Supplementary Fig. 12b). Interestingly, the addition of anti-CTLA-4 with either TGFβ1 or TGFβ3 inhibition, but not pan-TGFβ inhibition, had an even stronger effect in prolonging overall survival compared to isoform-specific TGFβ blockade alone but did not significantly affect tumor growth (Fig. 3 and Supplementary Fig. 12a). This discrepancy in overall survival may be attributable to the ability of CTLA-4 and TGFβ together to induce Treg development, leading to an immunosuppressive TME[46]. Isoform-specific TGFβ may circumvent this outcome by maintaining a predominantly tumor suppressor role. The combination of immune checkpoint blockade with isoform-specific TGFβ inhibition can produce additive results, as is seen with TGFβ and PD-1 inhibition and is not detrimental in combination with other immunotherapies.

In conclusion, anti-TGFβ therapy, either pan or isoform-specific, offers a means to counteract local immune resistance and has the potential to enhance responses to ICB. Through further exploration of the isoform-specific expression and function of TGFβ across cancer types, isoform-specific TGFβ inhibition offers a novel immunotherapeutic strategy to unleash the adaptive immune system against cancers that fail to respond to current checkpoint inhibitors.

## Methods

**Animal maintenance and ethics**. C57BL/6J and BALB/c mice were purchased from the Jackson Laboratory (Sacramento, CA). Animal experiments were performed in accordance with institutional guidelines under a protocol approved by the Memorial Sloan Kettering Cancer Center (MSKCC) Institutional Animal Care and Use Committee. All mice were maintained in a pathogen-free facility according to the National Institutes of Health Animal Care guidelines.

**Tumor cell lines**. The B16F10 mouse melanoma line was originally obtained from I. Fidler (MD Anderson Cancer Center, Houston, TX). The 4T1 mouse breast cancer cell line and CT26 colon carcinoma were purchased from ATCC (Manassas, VA). WG492 is a melanoma cell line derived from a tumor from the $BRAF^{V600E}$/ $PTEN^{-/-}$ transgenic mouse. These cells were maintained in RPMI 1640 containing 7.5% fetal bovine serum (FBS) and L-glutamine. Cells were detached using 0.25% trypsin/EDTA. For cell surface staining of tumor cells and $CD8^+$ T cell killing assays, cells were detached non-enzymatically using Cellstripper (Invitrogen).

**Histology and quantitative image analysis**. Eight to ten-weeks-old C57BL/6 female mice were implanted with 200,000 B16 cells or 200,000 WG492 cells

intradermally on the right flank; similarly, eight to ten-week-old BALB/c female mice were subcutaneously implanted with 200,000 4T1 cells or 200,000 CT26 cells on the right flank. Ten days post tumor challenge, tumors from mice that were euthanized with $CO_2$ were harvested and fixed in neutral buffered formalin for 48 h. Tissues were then processed in ethanol and xylene and embedded in paraffin in a Leica ASP6025 tissue processor. Paraffin blocks were sectioned at 5 microns, stained with hematoxylin and eosin (H&E), picrosirius red (PR), and an additional unstained section was used for IHC against alpha-smooth muscle actin (Abcam, ab32575). IHC was performed on a Leica Bond RX automated stainer using Bond reagents (Leica Biosystems, Buffalo Grove, IL), including a polymer detection system (DS9800, Novocastra Bond Polymer Refine Detection, Leica Biosystems). The chromogen was 3,3 diaminobenzidine tetrachloride (DAB), and sections were counterstained with hematoxylin. Cytoplasmic SMA expression on IHC and interstitial collagen content on picrosirius red was evaluated quantitatively by automated image analysis. Whole-slide digital images were generated on a scanner (Pannoramic 250 Flash III, 3DHistech, ×40/0.95NA objective, Budapest, Hungary) at a resolution of 0.121 μm per pixel. Image analysis was performed with HALO software Area Quantification module v.1.0 (Indica Labs, Albuquerque, NM). The region of interest (ROI) was manually defined as viable tumor tissue, excluding necrotic tumor tissue and adjacent non-tumor tissues. The area quantification module was used to detect the total amount of alpha-smooth muscle actin and collagen based on the optical density (OD) of DAB and picrosirius red staining, respectively. ROI selection, area quantification algorithm optimization, OD threshold determination, and validation of the results were performed by an ACVP board-certified veterinary pathologist (AOM).

**In vivo antibodies**. Anti-TGFβ1 (clone 13A1, mouse IgG1-k) was described previously[47] and anti-TGFβ3 (clone 1901 mouse IgG1-k) was obtained using the same procedure[28]. An ELISA was used to determine the binding specificities of the anti-TGFβ1 mAb clone 13A1 (IgG1) and anti-TGFβ3 mAb clone 1901 (IgG1) as previously described[47]. The specificity of these antibodies is shown in Supplementary Fig. 3a. These antibodies recognize only the active form of TGFβ and neutralize the binding of corresponding TGFβ ligands to their many receptors. Both antibodies were custom-ordered from Bioxcell. Anti-PD-1 (clone RMP1-14), anti-CTLA-4 (clone 9H10), anti-CD8 and anti-TGFβ1,2,3 (clone 1D11) were purchased from Bioxcell.

**In vivo mouse experiments and monoclonal antibody treatment**. Eight to ten-week-old C57BL/6 female mice were injected intradermally on the right hindlimb with 250,000 B16F10 cells in 50 microliters (50 μL) of phosphate-buffered saline (PBS). Similarly, 8 to 10-weeks-old BALB/c female mice were implanted with 500,000 CT26 cells in 50 μL of PBS subcutaneously injected on the right flank. On day 11, when tumors are 25–50 mm$^2$ in size, each mouse received intraperitoneal injections (i.p) of 0.2 mg of anti-TGFβ1, anti-TGFβ3, and anti-TGFβ1,2,3 (1D11) and every 2 days thereafter for a total of 8 doses (10 mice/group). In experiments where anti-PD-1 was given, each animal received 0.25 mg of anti-PD-1 (clone RMP1-14) i.p. in 0.2 mL of PBS starting on day 11 and every 3 days thereafter for a total of 5 doses. For experiments where anti-CTLA-4 was given, each animal received 0.1 mg of anti-CTLA-4 (clone 9H10) i.p. in 0.2 mL of PBS starting on day 11 and every 3 days thereafter for a total of 3 doses. For experiments where CD8$^+$ T cells were depleted, 0.2 mg of anti-mouse CD8$^+$ monoclonal antibody (clone 2.43) in 0.2 mL of PBS was given i.p. starting on day 11. Anti-CD8$^+$ therapy was given twice weekly for the duration of anti-TGFβ treatment. Each animal was tagged by ear notching on day 5 post tumor implantation and tumor size (length and width) was measured every 3–4 days using a caliper. Tumor measurements (surface area in mm$^2$) and overall survival were used to determine the efficacy of treatments.

For experiments where tissues were harvested for ex vivo analysis of immune infiltrates, 8 to 10-weeks-old C57BL/6 female mice were intradermally injected into the right flank with 500,000 B16F10 cells in 50 μL of PBS. On day 11, each animal received i.p. injections of 0.2 mg of anti-TGFβ1, anti-TGFβ3, and anti-TGFβ1,2,3 (1D11) and every 2 days thereafter for a total of 4 doses, and tumors and spleens were harvested 2 days after the last dose as detailed in the 'Flow cytometry analysis' section.

**Flow cytometry analysis**. C57BL/6 female mice were tumor-challenged and treated as described above. For flow cytometric analysis involving CT26 colon carcinoma, mice were tumor challenged as discussed. At different time points post tumor challenge when tumors were palpable, mice were euthanized and B16F10 tumors and corresponding spleens were harvested. Single-cell suspensions were prepared by mechanical dissociation through 40 μM cell strainers and red blood cells were removed from spleens using ACK lysis buffer (Lonza, Walkersville, MD). For staining for flow cytometry analysis: 100 μL of single-cell suspensions of each tissue were plated in 96-well round-bottom plates. Cells were pelleted by spinning at 2000 RPM for 5 min then incubated in 100 μL of 5 μg/mL Fc-block antibody (clone 2.4G2) for 20 min on ice in FACS buffer (PBS + 0.5% BSA + 2 mM EDTA). After Fc-block, cells were stained in FACS buffer containing fluorophore-conjugated surface antibodies and a fixable viability dye (efluor506, eBioscience) for 20 min on ice, then washed two times with 200 mL FACS buffer. All intracellular

staining was conducted using the Foxp3 fixation/permeabilization buffer according to the manufacturer's instructions (eBioscience). The samples were acquired on a LSRII (BD Biosciences) and data were analyzed using FlowJo software (version 10 - FlowJo, LLC). To characterize the cell-specific and temporal expression patterns of TGFβ isoforms on immune cells, Alexa Fluor 647-labeling of anti-TGFβ1 (clone 13A1) and TGFβ3 (clone 1901) was performed according to manufacturer instructions (Molecular Probes/Invitrogen) using succinimidyl ester dye. As our isoform-specific flow cytometry antibodies only recognize the active form of TGFβ, this suggests that most of the active ligand is bound to the surface of immune cells. Supplementary Fig. 13 shows the gating strategy used to classify different immune cells via flow cytometry analysis.

**Phospho-flow**. C57BL/6 female mice were tumor-challenged and treated as described above. Following four doses of anti-TGFβ treatment, tumors from treated animals (5 mice/group) were harvested and processed as described above for flow cytometric analysis. Following plating and pelleting of 100 μL of single-cell suspensions of each tumor tissue, cells were incubated with Fc-block antibody and a fixable viability dye for 20 min on ice in PBS. Cells were then washed two times with 200 μL PBS. Cells were then incubated overnight at 4 °C in the Foxp3 fixation/ permeabilization buffer solution, which was created according to the manufacturer's instructions (eBioscience). After 12 h, cells were stained in FACS buffer for 30 min on ice containing the following fluorophore-conjugated surface antibodies: CD45$^+$ Alexa Flour 700 (clone 30F-11), which had been tested to confirm staining of cells after fixation, and anti-Smad2 (pS465/pS467)/Smad3 (pS423/ pS425) PE (Clone O72-670, BD Biosciences), used according to the manufacturer's instructions. The samples were acquired on a LSRII (BD Biosciences) and data were analyzed using FlowJo software (version 10 - FlowJo, LLC).

**PrimeFlow RNA assay**. RNA detection by flow cytometry (PrimeFlow RNA Assay) was conducted to corroborate the detection of TGFβ isoforms by Alexa Fluor 647-labeled monoclonal antibodies. Gene-specific oligonucleotide target probes against non-homologous regions of TGFβ1 and TGFβ3 were created and pre-optimized by the manufacturer (ThermoFisher Scientific). A PrimeFlow kit containing necessary buffers and reagents was used according to the manufacture's protocol (ThermoFisher Scientific). Samples were stained with fluorescent-labeled antibodies against CD45$^+$, CD8$^+$, CD4$^+$, Foxp3$^+$, CD11b$^+$, CD11c$^+$, Ly6G$^+$, and Ly6C$^+$ to detect TGFβ protein and mRNA isoform expression on various immune cell subtypes. The samples were acquired on a LSRII and data were analyzed using FlowJo software as described above.

**CD8$^+$ T cell killing assay**. Single-cell suspensions of splenocytes were isolated from treated animals and CD8$^+$ T cells were purified using MACS beads according to the manufacture's protocol (Miltenyi). 1 mL complete RPMI medium containing 5 × 10$^5$ CD8$^+$ T cells and 10$^4$ B16F10 tumor cells (as targets) were added to 24-well plates and incubated for 48 h at 37 °C. 48 h later, the T cells were washed away with 1 mL PBS and the remaining viable tumor cells were detached using 0.25% trypsin/EDTA. The detached tumor cells were diluted and plated in six-well plates for colony formation. Seven days later, plates were fixed with 3.7% formaldehyde and stained with 2% methylene blue as previously described[48]. Colonies were counted manually to assess the number of viable tumor cells.

**IFNγ ELISpot assay**. Single-cell suspensions of splenocytes were isolated from treated animals and CD8+ T cells were purified using MACS beads according to the manufacture's protocol (Miltenyi). A mouse IFNγ ELISPOT kit was used and followed according to the manufacture's protocol (BD biosciences). Briefly, 1 × 10$^5$ CD8$^+$ T cells were co-cultured with 5 × 10$^4$ irradiated (60 Gy) B16F10 target cells for 16 h in a 96-well ImmunoSpot plate (Millipore). The plates were washed and processed according to the manufacture's protocol. DAB reagents were used for the detection of the IFN-γ spots (Vector Laboratories Inc). The spots were quantified using an ImmunoSpot S6 Micro Analyzer and ImmunoSpot Professional Software (Cellular Technology Limited).

**Quantitative PCR**. RNA from B16F10 cells maintained in-vitro and B16F10 tumors harvested 11 days post tumor challenge were purified using RNeasy Mini Kit according to the manufacture's protocol (Qiagen). Quantitative PCR was conducted using predesigned mouse Tgfb1, Tgfb2, and Tgfb3 Taqman probes (ThermoFisher) to determine the production of TGFβ isoforms by B16F10 cells in-vitro and whole tumor in vivo. Samples were acquired and analyzed using the Applied Biosystems 7500 Real-Time PCR System. 2^delta CT values were calculated relative to GAPDH expression.

**Statistics and reproducibility**. Unless otherwise indicated, p values were calculated using unpaired 2-tailed Student's t-test. A p-value of < 0.05 was considered statistically significant. For flow cytometry experiments, each experiment was repeated three times with 5 mice per group. As indicated in the corresponding figure legends, data are either representative of independent experiments or are displayed as fold change over the control after values from independent

experiments were pooled and normalized to corresponding control values. Significance between groups was determined using unpaired 2-tailed Student's t-test.

For functional T cell assays, including T cell killing assays and IFN-γ EliSpot, 3–5 mice per group were used and all experiments were repeated twice. For IHC experiments, 3–5 mice per group were used and representative cross sections are displayed in the figures.

For in vivo experiments, 10 mice per group were used and all in vivo experiments were repeated at least twice. All tumor growth curves were generated using Prism 9 software (GraphPad) and significant differences between groups were determined using 2-way ANOVA analysis. All tumor growth curves are plotted as mean ± standard error of the mean (SEM) and are representative of individual experiments. Overall survival curves were obtained from pooled replicate experiments and analyzed using Kaplan–Meier estimator. p-values comparing survival curves were calculated using the Log-rank (Mantel–Cox) test with $p < 0.05$ used for significance. All graphs and statistical calculations were done using Prism 9 software (GraphPad) and Microsoft Excel on MacOS. All statistically significant differences between groups are indicated on the figures.

**Reporting summary**. Further information on research design is available in the Nature Research Reporting Summary linked to this article.

## Data availability

All source data generated or analyzed during this study are included in this published article (and its Supplementary Information files as Supplementary Data 1). Additional data can be provided by the corresponding authors upon reasonable request.

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

## Acknowledgements
We are grateful to all the members of the Wolchok/Merghoub lab for their support. We are grateful for experimental support from the MSKCC Molecular Cytology Core Facility and the MSKCC Flow Cytometry Core Facility. This research was funded in part through the NIH NCI Cancer Center Support Grant (Immunology and Transplantation) P30 CA008748 54 (PI: Thompson; Program Co-Leader: Wolchok), NCI/NIH Immunization Against Melanoma Differentiation Antigens R01 CA056821 26 (PI: Wolchok), the Swim Across America, Ludwig Institute for Cancer Research, Parker Institute for Cancer Immunotherapy and Breast Cancer Research Foundation. In addition, the first author (A.G.) was supported by a Howard Hughes Medical Institute (HHMI) Research Fellows grant 2017–2018. R.G. acknowledges support from an NIH-T32 Postdoctoral Research Fellowship.

## Author contributions
The first author, A.G. formulated the project with the assistance of all listed authors. A.G. conceived, performed, and analyzed all experiments, with supervision from S.B. and assistance from R.G., K.F. and L.F.C. A.G. drafted and edited the manuscript with assistance from all listed authors. A.O.M. performed all histologic experiments and along with A.H. conducted corresponding quantitative image analysis. J.v.S., C.U. and G.R. provided flow cytometry and in vivo antibodies to conduct experiments. J.D.W. and T.M. helped to formulate the project, provided supervision, and edit the manuscript.

## Competing interests
J.D.W. is a paid consultant for Adaptive Biotech, Amgen, Apricity, Ascentage Pharma, Astellas, AstraZeneca, Bayer, Beigene, Bristol Myers Squibb, Celgene, Chugai, Eli Lilly, Elucida, F Star, Imvaq, Janssen, Kyowa Hakko Kirin, Linneaus, Merck, Neon Therapeutics; Novartis, Polynoma, Psioxus, Recepta, Takara Bio, Trieza, Truvax, Serametrix, Surface Oncology, Syndax, Syntalogic. J.D.W. receives research support from Bristol Myers Squibb, AstraZeneca, Sephora. J.D.W. has stock option ownership in Tizona Pharmaceuticals, Adaptive Biotechnologies, Imvaq, Beigene, Linneaus. T.M. is a paid consultant for Immunos Therapeutics, Pfizer Co-founder and Equity in IMVAQ therapeutics. T.M. receives research support from Bristol Myers Squibb, Surface Oncology, Kyn Therapeutics, Infinity Pharmaceuticals, Inc., Peregrine Pharmeceuticals, Inc., Adaptive Biotechnologies, Leap Therapeutics, Inc., Aprea. T.M. is an inventor on patent applications related to work on Oncolytic Viral therapy, Alpha Virus Based Vaccine, Neo Antigen Modeling, CD40, GITR, OX40, PD-1, and CTLA-4. The remaining author declares no competing interests.
