## [Transparent Peer Review File · Communications Biology]

Reviewers' comments:

Reviewer #1 (Remarks to the Author):

This work builds upon the previous discovery about the effects of TGF β on tumorigenesis and tumor suppress. By comparing the expression and production of TGF β in both stroma-heavy tumors and stroma-poor tumor, the authors supposed that targeting of isoform specific TGF β (TGF β 1 and TGF β 3) in the therapy of stroma-poor tumors may largely lower the off-target effect that elicited by total TGF β (pan-TGF β) neutralization. The topic of this manuscript has potential for a clinical translational apply, which will be an interesting and timely communication in the circle of stroma-poor cancer clinical application and research.

In reviewing of the current manuscript, the authors begin with their study from the observation of two murine tumor models, one is the B16 melanoma model stands for the stroma-poor cancer, the other one is the 4T1 breast cancer model stands for the stroma-heavy cancer. By comparing of the picosirius red (PR) staining and the expression of alpha-smooth muscle actin (α SMA), the authors primarily found that different cancer may have different stroma microenvironment for cancer supplying. Then, the authors focused on the isoform specific expression of TGF β in B16 melanoma tumor model. Respectively identified TGF β 1 and TGF β 3 have the highest expression in/on tumor infiltrating myeloid immune cells and dendritic cells (DCs). In vivo test has indicated that singly administration of pan-TGF β Ab and isoform specific TGF β 1 and TGF β 3 Abs will significantly delay the B16 melanoma growth. Further immunologic analysis found that isoform specific TGF β inhibition contributes to the proliferation and activation of CD8+ T effector cells in the TME. To verify the critical effect of CD8+ T cells on anti-tumor function of TGF β inhibition, the authors artificially depleted the CD8+ T cells by treatment with anti-CD8 Ab, and found that depletion of CD8+ T cells will abrogate anti-tumor effect of TGF β inhibition. Further in vivo test by combination with immune checkpoint-based immunotherapy, the authors declared that TGF β 1 inhibition will synergistically contributes to the effect of immune checkpoint-based immunotherapy on B16 melanoma. CTLA-4 and PD-1/PD-L1 as the most popular immune checkpoints, now has been clinically used as the standard option to treat melanoma and other cancers. However, as the authors have indicated in the manuscript, not all patients have robust and durable responses to these therapies. With the aim of improving the therapeutic effect of immune checkpoint blockade (ICB), the authors identified isoform specific TGF β as the synergistic ICB target to reduce the "off-target" effect by blocking total TGF β , which may elicit a significant contribution to the precision medicine in the near future.

Taken together, novelty of this work is suitable and the data is well presented, the experiments are well planned, and the paper overall is well thought out. However, there is still a long way to be accepted and published. In the spirit of making this work better, I sincerely provide my advices to the authors.

I have some criticisms as shown below:

Major ones:

1. As one of the most important basestones of this topic, data in figure1 looks too weak, to strengthen of which the authors should provide two kinds of such clinical tumor samples to analysis and statistic. To be better address this issue, the authors should respectively give at least two kinds of stroma-poor and stroma-heavy murine cancer models with significant difference of PR staining and α SMA expression.
2. After creating single cell suspensions from B16 tumor, the single cells were subjected to flow cytometry analysis, expression of isoform specific TGF β were compared in/on kinds of critical immune cells, however, as one of the major inhibitory immune cells in the TME, the myeloid-derived suppressor cells (MDSC) haven't been mentioned (Figure 2a and Suppl Figure 3a), but why?

3. As the authors have indicated in Figure 4, isoform specific TGF β inhibition induces CD8+ T cell activation and proliferation, however, it was believed that other tumor-infiltrated immune cells may also have changed, such as the composition of Tregs and MDSCs, as well as NKs and effector CD4+ T cells and so on. What's more, effector cytokines and cytotoxic agents such as IFN γ and perforin may also need to be monitored. Single GranzymeB expression by CD8+ T cells may be not comprehensively characterized its functional status, CD107a and Granzyme B double positive CD8+ T cells may be better characterized its functional status.

4. Data in Figure 3 to Figure 6 and related data in suppl. Figure 6 to suppl. Figure 9, the comparisons between α TGF β 1 or α TGF β 3 treated group and 1D11 treated group also need to be made. In addition, as a positive control of the in vivo test, matched isotype Ig treated group also need to be contained in such as Figure 6 and Supply. Figure 9.

Also, there are some minor mistakes:

5. In line 254 (Delete word "old"), line 432(Change "and" to "or") and line 467 (Delete the second "day").

6. The figure legend in Suppl. Figure 1 is missed.

7. The microliter or microgram should be labeled as " μ L" or " μ g", rather than "uL" or "ug".

8. CD8 positive T cells should be consistently labeled as CD8+ T cells.

9. The ratio of effector to target cell (50:1) looks not very make sense. Does the author have a dilution test of the ratio? Could the author provide related data to support this ratio?

10. Some statistic comparison between groups are lost, such as in Figure 4f, Suppl. Figure 6c and Suppl. Figure 7. Overall, the statistic should be carefully reorganized, the clearer the better.

11. The tumor size calculated as volume (mm³) should be better than calculated as area (mm²) in solid tumor, could the authors explain about this?

Reviewer #2 (Remarks to the Author):

In this manuscript, Dr. Gupta and colleagues show that targeting isoform-specific TGF-beta have better anti-tumor response than pan-TGF-beta inhibition. Mechanistically, targeting TGF-beta isoform enhance CD8 T cell function. Moreover, they showed that combining TGF β inhibition with immune checkpoint blockade results in improved tumor control.

Overall, this is an interesting work with clear translational implication for immunotherapy in stroma poor tumors. I only have some minor comments

1 - Some key references should be included. For instance, in this sentence "modulate the extracellular environment (ECM) and decrease immune surveillance, leading to metastasis and treatment resistance" these articles should be cited: - A. Calon, et al.

Stromal gene expression defines poor-prognosis subtypes in colorectal cancer

Nat. Genet., 47 (2015), pp. 320-329, - Chakravarthy, A et al. TGF- β -associated extracellular matrix genes link cancer-associated fibroblasts to immune evasion and immunotherapy failure. Nat Commun 9, 4692 (2018) and - S. Mariathasan, et al.

TGFbeta attenuates tumour response to PD-L1 blockade by contributing to exclusion of T cells
Nature, 554 (2018), pp. 544-548

2 -It is acceptable to use only one model to study the detailed mechanism, but having at least one more stroma poor model to show the anti-tumor effect (Figures 3 and 6) would increase the generalization of these findings.

3 - In figure 4 panels A to D suggest an hierarchy of effect on CD8 T cell function:
1D11>aTGFB3>aTGFB1. However, panel E-F as well as Figure 3 suggests almost the opposite
1D11<aTGFB3<aTGFB1. Could the authors discuss this discrepancy?

4 - How statistics were calculated in Figure 6? What does the error bar represent (SEM? SD? Else?)?
Some P values are very small, while the difference seems to be small and the error bars large. In general, it would help to have the statistical test and the error bar defined in the figure legends.

We thank the reviewers for their careful and well thought out analyses and critiques of the manuscript. We have addressed the reviewers' comments below to the best of our ability in a point-by-point manner. Our responses are indicated in light blue font. We have also incorporated figures in our response below. Figures in the point by point response are referred to as figure R.

Reviewer #1

1. As one of the most important basestones of this topic, data in figure1 looks too weak, to strengthen of which the authors should provide two kinds of such clinical tumor samples to analysis and statistic. To be better address this issue, the authors should respectively give at least two kinds of stroma-poor and stroma-heavy murine cancer models with significant difference of PR staining and α SMA expression.

We know have incorporated two models of stroma poor (B16 and CT26) and stroma enriched (4T1 and WG492) cell lines in our study. After analyzing the stroma of multiple tumor types (data not shown), we chose 2 additional tumors CT26 (colon carcinoma) and WG492 (Braf^{V600E}PTEN^{-/-} melanoma) as models of stroma-poor and stroma-rich murine cancers, respectively. As shown in figure R1 (included as supplementary figure 2 in the revised manuscript and discussed on page 17 starting line 451), we found that compared to WG492 melanoma, CT26 colon has less picosirus red staining and immunoreactivity against α SMA (figure R1). As picosirus red stains collagen fibers and α SMA is a surrogate marker to identify fibroblasts, this suggests that CT26, like B16 melanoma, is a stroma poor tumor model with fewer collagen fibers and fibroblast-like cells. This suggests that the primary source of TGF β expression in CT26 colon may be from infiltrating immune cells and not local stroma or fibroblasts as with B16 melanoma (figure 1, 2). We have updated the methods section (pages 8-15) to reflect our use of CT26 colon carcinoma. We subsequently used CT26 as another stroma poor tumor model to investigate the efficacy of isoform specific TGF β inhibition on tumor growth (see reviewer #2 question 2 and Figure 3d).

Figure R1: CT26 colon and WG492 melanoma as examples of stroma poor and stroma heavy murine tumor models, respectively

Syngeneic mice were implanted with 200,000 CT26 and 200,000 WG492 cells injected subcutaneously (n = 3 mice/group). Tumors were harvested 10 days post tumor challenge and fixed for immunohistochemistry (IHC) prior to staining with picosirus red (PR) and alpha-smooth muscle actin (α SMA). a) Representative cross sections of CT26 colon (top) and WG492 melanoma (bottom) stained for α SMA and PR. b) Bar graphs demonstrate quantification of the staining of either PR or α SMA \pm standard error of the mean (SEM) following analysis by Halo software with supervision from a pathologist. **p<0.005

2. After creating single cell suspensions from B16 tumor, the single cells were subjected to flow cytometry analysis, expression of isoform specific TGF β were compared in/on kinds of critical immune cells, however, as one of the major inhibitory immune cells in the TME, the myeloid-derived suppressor cells (MDSC) haven't been mentioned (Figure 2a and Suppl Figure 3a), but why?

The data in supplementary figure 9 shows that there were no significant changes in the frequencies of neutrophils (granulocytes) or monocytes in the tumor microenvironment. The phenotypic markers for neutrophils (CD11b+ Ly6G+) are the same marker for granulocytic MDSCs (G-MDSCs). Similarly, the phenotypic markers for monocytes (CD11b+ Ly6C^{hi}) are also the same markers for monocytic MDSCs (M-MDSCs). However, we do not refer to these cells as MDSCs because we have tested these cells functionally and have shown that, in B16 melanoma, these cells are not immunosuppressive when compared to other tumor models enriched with MDSCs.^{1,2} We have amended the manuscript (page 19, lines 496-499) to include this explanation as to why MDSCs were not analyzed.

3. As the authors have indicated in Figure 4, isoform specific TGF β inhibition induces CD8+ T cell activation and proliferation, however, it was believed that other tumor-infiltrated immune cells may also have changed, such as the composition of Tregs and MDSCs, as well as NKs and effector CD4+ T cells and so on. What's more, effector cytokines and cytotoxic agents such as IFN γ and perforin may also need to be monitored. Single GranzymeB expression by CD8+ T cells may be not comprehensively characterized its functional status, CD107a and Granzyme B double positive CD8+ T cells may be better characterized its functional status.

In the response to this question we have further detailed the analysis of Tregs and MDSCs, as well as NK cells and effector CD4+ T cells. We have also detailed the effector functions of the T cells below.

We have analyzed the immune infiltrate in tumors treated with 4 doses of either anti-TGF β 1, anti-TGF β 3 and pan-TGF β inhibition, 1D11. We did not find significant changes in the composition of the immune microenvironment with regards to myeloid/ dendritic cells (supplementary fig 9) or other lymphoid cell types. As shown in supplementary figure 9, there were no substantial changes in the quantities of granulocytes, monocytes or macrophages (figure R2a, see response to comment 2 regarding MDSCs) and upon further investigation, there were no changes in the polarization status of M1 (MHCII+, CD206-) macrophages relative to the control (figure R2b). However, there was a significant decrease in M2 (MHCII-, CD206+) macrophages in the anti-TGF β 3 and 1D11 treated groups as shown in figure R2c. No difference in the ratio of M1/M2 macrophages was detected across the treatment groups compared to untreated controls (figure R2d). To satisfy the reviewer comment, we have now included this data as an additional supplemental figure in the revised manuscript (page 21, lines 550-553) as supplementary figure 10.

Figure R2: Isoform specific TGF β inhibition has no effects on tumor infiltrating macrophage populations

B16 tumors were implanted into mice and harvested according to the scheme shown in Figure 4a. Tumors underwent processing for flow cytometry and analysis as discussed in Materials and Methods section. a) Plot showing F4/80+ macrophages harvested from animals treated with isoform specific versus pan-TGF β inhibition. Gating was first done on CD45+ immune cells, followed by CD11b+ (excluding CD3+ T cells) and then F4/80+ macrophages. b) Plot showing M1 macrophage subpopulation defined as MHCII+, CD206-. c) Plot showing M2 macrophage subpopulation defined as MHCII-, CD206+. d) Plot shows the ratio of M1/M2 macrophages in each group of treated mice. Data represents pooled values from two independent experiments normalized to the control (n = 5 mice/ group) and is displayed as fold change compared to the control \pm SD gated on indicated cells types. *p<0.05; **p<0.001; ***p<0.0005

Regarding the quantities of other tumor infiltrating immune cells, there were no significant changes in the CD4+ T effector (Foxp3-) or CD4+ Treg (Foxp3+) populations across treatment groups (figure R3a, b). 1D11 did demonstrate an increase in Tregs, but no significant changes were seen among the animals treated with isoform specific anti-TGF β inhibition (figure R3b). There was a significant decrease in NK cells among animals treated with anti-TGF β 3 and 1D11 compared to both control and anti-TGF β 1, suggesting a possible mechanism of NK cell infiltration mediated by TGF β 3 (figure R3c). Furthermore, no significant differences were detected in the quantities or activation status of CD4+ T effector cells (Foxp3-) or CD4+ Tregs (Foxp3+) as shown in figure R3d, e. These findings are discussed on page 21, line 549 of the edited manuscript.

Figure R3: Isoform specific TGFβ inhibition has limited effects on CD4+ T cells

B16 tumors were implanted into mice and harvested according to the scheme shown in Figure 4a. Tumors underwent processing for flow cytometry and analysis as discussed in Materials and Methods section. Plots showing quantities of tumor infiltrating a) CD4+ T effector cells, b) CD4+ Tregs, and c) NK cells. Plots showing Ki67+ and PD1+ expression on tumor infiltrating d) CD4+ T effectors and e) CD4+ Tregs. * $p < 0.05$; ** $p < 0.001$

Further analysis using additional CD8+ activation markers in the tumors of treated animals did not demonstrate significant trends in CD8+ T cell phenotypes among isoform specific or pan-TGFβ inhibition. In particular, there were no significant differences among CD8+ T cells that make IFNγ following re-stimulation ex vivo with PMA and ionomycin (figure R4a). In addition, we did not observe any dramatic changes in CD107α+ or GranzymeB+CD107α (as suggested by the reviewer) among anti-TGFβ groups as shown in figure R4b, c. Anti-TGFβ1 and 1D11 did demonstrate a significant increase in perforin expression among CD8+ T cells compared to controls (figure R4d), which is similar to what we observed with Granzyme B expression in figure 4d with the exception of increased cytolytic granule expression in the anti-TGFβ3 treatment group. While expression of these proteins are all surrogate markers of CD8+ T cells' killing potential, the most direct measurement of CD8+ T cell effector function is the killing activity of these cells shown in Figure 4e.

Figure R4: Cytokine expression by tumor infiltrating CD8+ T cells did not differ among groups treated with isoform specific TGFβ inhibition

B16 tumors were implanted into mice and harvested according to the scheme shown in Figure 4a. Tumors underwent processing for flow cytometry and analysis as discussed in Materials and Methods section. For panels a-c, CD8+ T cells were harvested ex-vivo and then restimulated with PMA/ ionomycin. a) Plot showing restimulated CD8+ T cells expressing IFN γ . b) Plot showing restimulated CD8+ T cells expressing CD107 α . c) Plot showing restimulated CD8+ T cells that co-expressed Granzyme B and CD107 α . d) Plot showing unstimulated CD8+ T cells expressing the cytotoxic enzyme perforin.

4. Data in Figure 3 to Figure 6 and related data in suppl. Figure 6 to suppl. Figure 9, the comparisons between α TGF β 1 or α TGF β 3 treated group and 1D11 treated group also need to be made. In addition, as a positive control of the in vivo test, matched isotype Ig treated group also need to be contained in such as Figure 6 and Supply. Figure 9.

We thank the reviewer for his or her insight and below have detailed additional statistical analyses. All updated figures are included in the resubmission.

- For Figure 3c, all statistically significant differences between untreated animals and those treated with α TGF β 1, α TGF β 3 or 1D11 are indicated on the original figure. Only α TGF β 3 demonstrated a superior anti-tumor effect in comparison to 1D11 with $p < 0.05$ in the B16 model. Neither α TGF β 1 versus 1D11 nor either of the isoform specific anti-TGF β therapies demonstrated improved growth control over the other in B16. In CT26, all significant differences between groups is illustrated on figure 3d.
- For Figure 4b, only α TGF β 3 versus control and 1D11 versus control demonstrated significant increases in the CD45+ immune infiltrate and CD8+ tumor infiltrating lymphocyte (TIL). No significant differences were seen with regards to the CD45+ and CD8+ immune infiltrate amongst the other treatment groups. Similarly, for all CD8+ T cell phenotypes shown in Figure 4d, only the statistically significant differences are illustrated. With the exception of Granzyme B staining in which α TGF β 1 treated animals showed greater Granzyme B expression compared to control, only α TGF β 3 and 1D11 groups demonstrated increased expression of the listed markers. There were no statistically significant differences seen between the other treated groups.
- Figure 4e has been updated to reflect significant differences between each treatment group. Not only did isoform specific α TGF β therapy result in enhanced CD8+ T cell killing when compared to untreated animals, but also produced significantly greater

killing in comparison to 1D11. Further analysis revealed that CD8+ T cells harvested from animals treated with α TGF β 3 demonstrated superior killing against B16 cells when compared to CD8+ T cells harvested from untreated animals and those treated with α TGF β 1 or 1D11. α TGF β 3 treatment resulted in the greatest statistically significant killing of B16 melanoma cells in-vitro when compared to either α TGF β 1 or 1D11 treatment.

- Figure 4f has also been updated to reflect significant differences between each treatment group. Not only did α TGF β 1 treatment result in increased IFN- γ production by CD8+ T cells when compared to the untreated group, but also demonstrated significantly increased antigen-specific cytokine responses from CD8+ T cells compared to the 1D11 group.
- Figure 5a has been updated to demonstrate statistically significant depletion of CD8+ T cells in both the spleen and tumor. The first panel in Figure 5b (top left) has also been updated to reflect the superior anti-tumor effect of isoform specific TGF β inhibition in comparison to 1D11. α TGF β 1 and α TGF β 3 demonstrated statistically significant tumor control when compared to 1D11.
- Figure 6 includes all significant differences between treated groups. Differences that were not statistically significant were omitted to minimize the busyness of the figure.

Supplementary figures:

- Supplementary figure 8c (formally supplementary 6c) has been updated to include the statistically significant decrease in pSMAD expression following isoform specific α TGF β therapy in comparison to 1D11. Both α TGF β 1 and α TGF β 3 resulted in a greater reduction in pSMAD expression on CD45+ immune cells compared to pan-TGF β inhibition.
- The statistics of supplementary figure 9 (formally supplementary 7) have been revised as well. The most prominent finding was the significant decrease in CD11c+ dendritic cells among animals treated with α TGF β 3 or pan-TGF β inhibition.
- The statistics of supplementary figure 11 (formally supplementary 8) have also been revised to include the significant reduction in PD-L1 expression on CD11c+CD8+ dendritic cells from animals treated with 1D11 compared to α TGF β 3.
- Revision of supplementary figure 12 (formally supplementary figure 9) further revealed that the addition of α CTLA4 treatment did not improve survival over that achieved with α CTLA4 treatment monotherapy; however, it once again demonstrated that the addition of α PD1 did increase survival over that achieved by α TGF β inhibition alone.

We found that treatment with an IgGk1 (α TGF β isotype control) produced comparable tumor growth curves to control (untreated) animals as shown below in figure R5a. Our lab and others have shown that mice bearing B16 tumors do not respond to immune checkpoint monotherapy with α CTLA-4 or α PD-1³. As such, we did not expect that single agent blockade by either anti-CTLA-4 or anti-PD-1 to produce significant tumor control. As shown below in figure R5b-d there were no significant differences in tumor growth or overall survival among control animals and those treated with anti-CTLA-4 versus isotype control (hamster IgG) or anti-PD-1 versus isotype control (rat IgG2a). Moreover, three experiments comparing anti-CTLA-4 or anti-PD-1 monotherapy against B16 tumor growth resulted in similar growth curves as shown in figure R6.

Figure R5: Monotherapy with checkpoint inhibitors does not provide tumor control in B16 melanoma
 B16 tumors were implanted into mice and harvested as discussed in Materials and Methods a) Growth curves for control animals and those treated with IgGk1 (α TGF β isotype control) showing no significant differences in tumor growth. b) Individual tumor growth curves for animals treated with control, anti-CTLA4 and anti-PD-1. c) Plot showing overall tumor growth curves for control, anti-CTLA4, hamster IgG (anti-CTLA4 isotype control), anti-PD-1 and rat IgG2a (anti-PD-1 isotype control). No significant differences were found between tumor growth curves. d) Survival curves for aforementioned treatment groups. No significant differences were seen among the tumor growth or survival curves.

Figure R6: Three experiments comparing single agent checkpoint blockade shows limited tumor control in B16 melanoma
 B16 tumors were implanted into mice and harvested as discussed in Materials and Methods. In each experiment (exp), animals were treated with control, anti-CTLA4 and anti-PD1. Across all three experiments, similar tumor growth curves were derived with limited tumor control against B16 melanoma.

Also, there are some minor mistakes:

5. In line 254 (Delete word “old”), line 432(Change “and” to “or”) and line 467 (Delete the second “day”).

We thank the authors for this comment and the appropriate changes to the manuscript have been made.

6. The figure legend in Suppl. Figure 1 is missed.

We apologize for the inconvenience and have assured that the figure legend for supplemental figure 1 is present.

7. The microliter or microgram should be labeled as “ μL ” or “ μg ”, rather than “uL” or “ug”.

We thank the authors for this comment and have ensured consistent labeling of microliter or microgram throughout the manuscript.

8. CD8 positive T cells should be consistently labeled as CD8+ T cells.

We thank the reviewers for this comment and have ensured consistent labeling of CD8+ T cells throughout the manuscript.

9. The ratio of effector to target cell (50:1) looks not very make sense. Does the author have a dilution test of the ratio? Could the author provide related data to support this ratio?

The 50:1 effector to target ratio used in this experiment is due to the fact that we used CD8+ T cells isolated from the spleen and not the tumor. It is technically difficult to conduct a killing assay using CD8+ T cells isolated from the tumors as the tumors from the treated groups are too small to isolate enough CD8+ T cells for a killing assay. The conditions used for killing assay in Figure 4e were specifically designed and optimized to stringently test the differences in the cytolytic ability of CD8+ T cells isolated from the spleen of treated animals. We have used similar methods and experimental conditions in prior publications^{4,5}.

10. Some statistic comparison between groups are lost, such as in Figure 4f, Suppl. Figure 6c and Suppl. Figure 7. Overall, the statistic should be carefully reorganized, the clearer the better.

We thank the reviewers for their input. For each figure and supplemental figure significant statistical comparisons between groups are included. Figure legends for each figure and supplemental figure have also been updated to more clearly describe the displayed statistical analysis. The methods section of the manuscript was also modified to better elucidate the statistical analyses utilized (pages 8-15).

11. The tumor size calculated as volume (mm^3) should be better than calculated as area (mm^2) in solid tumor, could the authors explain about this?

As we aimed to assess an orthotopic tumor response, B16 cells were injected intra-dermally into the right flank of mice. We found that as the melanoma tumors grew, they expanded laterally over vertically. Therefore, our initial measurements showed that the height of the tumors were negligible compared to the length and width. As a result, we measured length and width and calculated tumor area (mm^2 using the equation of a circle or ellipse πr^2) in order to more accurately and consistently monitor the size of melanoma tumors. Measuring tumor growth as surface area versus volume (using the commonly used formula of $(\text{length})(\text{width}^2)(0.52)$) produces similar tumor growth curves and statistical relationships (figure R7).

Figure R7: No difference in tumor growth curves when plotted as surface area versus volume
 Figure 3a demonstrates the treatment paradigm using isoform specific anti-TGF β therapy. a) Overall tumor growth curves measured as surface area (length x width x π) shown for untreated animals and animals treated with anti-TGF β 1, anti-TGF β 3, and 1D11 (pan-TGF β inhibition). Data is presented \pm standard error of the mean (SEM). b) Overall tumor growth curves measured as volume (length x width² x 0.52) for the aforementioned untreated and treated groups. Growth curves demonstrate a similar pattern as when measured using surface area. Data is presented \pm SEM. Statistics were calculated 24 days post tumor implantation. *p<0.05; ***p<0.01; ****p<0.0005; *****p<0.0001.

Reviewer #2 (Remarks to the Author):

In this manuscript, Dr. Gupta and colleagues show that targeting isoform-specific TGF-beta have better anti-tumor response than pan-TGF-beta inhibition. Mechanistically, targeting TGF-beta isoform enhance CD8 T cell function. Moreover, they showed that combining TGF β inhibition with immune checkpoint blockade results in improved tumor control.

Overall, this is an interesting work with clear translational implication for immunotherapy in stroma poor tumors. I only have some minor comments

1 - Some key references should be included. For instance, in this sentence “modulate the extracellular environment (ECM) and decrease immune surveillance, leading to metastasis and treatment resistance” these articles should be cited: - A. Calon, et al. Stromal gene expression defines poor-prognosis subtypes in colorectal cancer Nat. Genet., 47 (2015), pp. 320-329, - Chakravarthy, A et al. TGF- β -associated extracellular matrix genes link cancer-associated fibroblasts to immune evasion and immunotherapy failure. Nat Commun 9, 4692 (2018) and - S. Mariathasan, et al. TGFbeta attenuates tumour response to PD-L1 blockade by contributing to exclusion of T cells Nature, 554 (2018), pp. 544-548

We thank the reviewer for his/her suggestions and have included the requested citations.

2 –It is acceptable to use only one model to study the detailed mechanism, but having at least one more stroma poor model to show the anti-tumor effect (Figures 3 and 6) would increase the generalization of these findings.

Based off our IHC analysis illustrating CT26 as a representative stroma-poor model (see answer to reviewer #1 question 1 and supplemental figure 2), we assessed the efficacy of in-vivo isoform specific TGF β inhibition in another stroma-poor cancer type. We first characterized the expression of TGF β isoforms on infiltrating immune cells as was done for B16 melanoma shown

in figure 2. Relative to the immune microenvironment of B16, we found greater expression of TGF β 1 and TGF β 3 on infiltrating immune cells as quantified by MFI (figure R8a). In CT26 the predominant isoform of TGF β expressed on immune cells is TGF β 1 as seen by higher detected MFI levels of TGF β 1 compared to TGF β 3. Similar to B16 melanoma both isoforms are primarily expressed by the myeloid-dendritic cell population (figure R8a, figure 2).

Isoform specific inhibition of TGF β demonstrated superior tumor control with anti-TGF β 1 compared to anti-TGF β 3 in CT26 (figure R8b). While in B16 there appears to be equal expression of both isoforms on infiltrating immune cells (figure 2, figure R8a) as seen by equivalent MFI values of both isoforms, in CT26 the TGF β signature of the tumor microenvironment is mainly defined by TGF β 1 expression. Without ample ligand to inhibit in CT26, TGF β 3 inhibition is unable to suppress tumor growth while TGF β 1 inhibition significantly delayed tumor progression (figure R8b). In B16, with relatively high expression of both TGF β 1 and TGF β 3, isoform specific inhibition with either anti-TGF β 1 and anti-TGF β 3 produced significant tumor control (figure R8a, b). These results illustrate that each stroma-poor tumor type has a specific TGF β signature with different balances of TGF β 1 versus TGF β 3 in the local microenvironment⁶. As a result, inhibition of one isoform of TGF β versus another may produce differential effects on tumor growth based on the local expression of the predominant TGF β isoform. Canè et al showed similar results with TGF β 1 inhibition potentiating the anti-tumor effect of prophylactic vaccination with irradiated CT26 cells⁷ and Terabe et al demonstrated that inhibition of TGF β 1 and TGF β 2 can reduce tumor burden in lungs with a metastatic CT26 model⁸. Our results illustrating high expression of TGF β 1 in CT26 tumors coupled with in-vivo efficacy data and previously published studies establish CT26 as having a TGF β 1 “signature” responsive to TGF β 1 inhibition. Furthermore, using the TiRP model of autochthonous melanoma, Canè et al demonstrated that TiRP melanoma is characterized by high expression of both TGF β 1 and TGF β 3 transcripts which they found are primarily produced by the tumor cells and “stroma” (defined as non-tumor cells), respectively.⁷ Furthermore, a recent paper by Martin et al demonstrated that TGF β isoform expression varies across tumor types based on mRNA expression. Using RNA-sequencing data from The Cancer Genome Atlas they found that while TGF β 1 mRNA is the most prevalent isoform expressed in the majority of human cancers, certain cancer types, such as breast, mesothelioma and prostate, are defined by high expression of both TGF β 1 and TGF β 3 mRNA.⁶ Analysis of skin cutaneous melanoma showed expression of both isoforms with higher expression of TGF β 1 mRNA.⁶

The results of Martin et al suggest that each tumor type has a specific TGF β signature. Our analysis demonstrates that CT26 and B16, both of which are stroma-poor murine models (figure 1, figure R1), have differential expression of TGF β isoforms by infiltrating immune cells with CT26 tumors expressing greater amounts of TGF β 1 compared to high expression of both isoforms in B16 (figure R8a). We have recapitulated the anti-tumor effect of isoform specific TGF β inhibition in another stroma-poor model by achieving delayed tumor growth via inhibiting the predominant TGF β isoform in CT26 using TGF β 1 blockade (figure R8b). We have included these results as supplementary figure 6 and revised figure 3d in the updated manuscript (page 19, lines 500-505 and page 20, lines 531-539). We have also included these results in the discussion section of the updated manuscript (page 25, line 640-662).

Figure R8: Isoform specific inhibition in CT26, another stroma-poor murine cancer model

200,000 CT26 tumor cells were implanted subcutaneously and 250,000 B16 cells were implanted intradermally into the right hind flank of syngeneic mice. Tumors were harvested and processed as discussed in Materials and Methods for flow cytometry analysis. a) Plots showing relative MFI expression of TGFβ1 and TGFβ3 on infiltrating immune cells in CT26 (top) and B16 (bottom) tumors compared to the negative stain or fluorescence minus one (FMO). b) Tumor growth curves of CT26 (top) and B16 (bottom) treated with anti-TGFβ1, anti-TGFβ3 and 1D11 (pan-TGFβ inhibition). * $p < 0.05$; *** $p < 0.0005$; **** $p < 0.0001$.

3 - In figure 4 panels A to D suggest an hierarchy of effect on CD8 T cell function: 1D11 > αTGFβ3 > αTGFβ1. However, panel E-F as well as Figure 3 suggests almost the opposite 1D11 < αTGFβ3 < αTGFβ1. Could the authors discuss this discrepancy?

Figures 4b-d describes the composition of the immune infiltrate in tumors treated with pan or isotype specific TGFβ inhibition. While CD8+ T cells harvested from animals treated with 1D11 may display higher levels of proliferation (Ki67+) or cytotoxic ability (Granzyme B+), this does not necessarily correlate with greater functional status. In addition, the phenotypic expression shown through activation markers such as Ki67+, Granzyme B+ or PD-1+ may not directly correlate with the cytotoxic effects of CD8+ T cells. Figures 3 and 4e and 4f are functional studies that demonstrate the cytotoxic anti-tumor effects of CD8+ T cells. The most striking finding is that the *in-vivo* anti-tumor effect of αTGFβ3 treatment shown in figure 3 correlates with the cytotoxic effect of CD8+ T cells as shown in the *in-vitro* killing assay in figure 4e. These CD8+ T cells displayed superior cytotoxic effects compared to all other treatment groups. Lastly, figure 4f shows that isotype specific anti-TGFβ treatment and in particular αTGFβ1 treatment resulted in antigen specific CD8+ T cells that can secrete an effector cytokine, IFNγ, in response to the tumor *ex vivo*. We do not observe a direct correlation between cytokine production and killing potential of CD8+ T cells or anti-tumor efficacy.

It has been previously described that the T cells that make cytokines such as IFNγ are distinct from T cells that are cytolytic⁹. The difference in cytolytic (figure 4e) and antigen-specific (figure 4f) CD8+ T cell responses may be due to isotype specific effects of anti-TGFβ1 versus anti-TGFβ3. Recent data demonstrated that OVA-CD8+ T cells with ALK5 loss (a component of the TGFβ receptor and mimics pan-TGFβ inhibition) had higher levels of

cytotoxic killing against an OVA-expressing tumor line as well as higher IFN γ production compared to wildtype CD8 $^+$ T cells¹⁰. These results are similar to those seen with 1D11 leading to higher rates of cytotoxic and antigen specific responses (figure 4e, 4d). Another study showed that CD8 $^+$ T cells that express latent associated peptide (LAP $^+$), which forms the inactive latent TGF β complex and curtails the effect of all isotypes of active TGF β , express high levels of IFN γ ¹¹. However, all recent studies have evaluated killing potential and IFN γ production in the setting of pan-TGF β inhibition. As TGF β has multiple modulators and receptors all with differing affinities for the various TGF β isotypes, it is possible that inhibition of TGF β 1 versus TGF β 3 has differing effects on CD8 $^+$ cytotoxicity versus antigen specificity.

In conclusion, these figures suggest that the greatest correlation with B16 tumor control is the killing ability of CD8 $^+$ T cells isolated from animals treated with α TGF β 3 showing superior response in these aspects.

4 - How statistics were calculated in Figure 6? What does the error bar represent (SEM? SD? Else?)? Some P values are very small, while the difference seems to be small and the error bars large. In general, it would help to have the statistical test and the error bar defined in the figure legends.

We appreciate the insight provided by the reviewer. We have updated the figure legend of figure 6 to explain that the data is shown standard error of the mean (SEM). We have updated the figure legends for all figures, including supplemental figures, and the methods section of the manuscript to more clearly describe the statistical analysis conducted. For all growth curves displayed in the manuscript (fig 3, fig 5, fig 6), significant differences between groups was determined using 2 way ANOVA analysis and data is displayed as mean \pm SEM.

References

- 1 De Henau, O. *et al.* Overcoming resistance to checkpoint blockade therapy by targeting PI3Kgamma in myeloid cells. *Nature* **539**, 443-447, doi:10.1038/nature20554 (2016).
- 2 Holmgaard, R. B. *et al.* Tumor-Expressed IDO Recruits and Activates MDSCs in a Treg-Dependent Manner. *Cell Rep* **13**, 412-424, doi:10.1016/j.celrep.2015.08.077 (2015).
- 3 Spranger, S. *et al.* Mechanism of tumor rejection with doublets of CTLA-4, PD-1/PD-L1, or IDO blockade involves restored IL-2 production and proliferation of CD8(+) T cells directly within the tumor microenvironment. *J Immunother Cancer* **2**, 3, doi:10.1186/2051-1426-2-3 (2014).
- 4 Budhu, S. *et al.* Targeting Phosphatidylserine Enhances the Anti-tumor Response to Tumor-Directed Radiation Therapy in a Preclinical Model of Melanoma. *Cell Rep* **34**, 108620, doi:10.1016/j.celrep.2020.108620 (2021).
- 5 Khalil, D. N. *et al.* In situ vaccination with defined factors overcomes T cell exhaustion in distant tumors. *J Clin Invest* **129**, 3435-3447, doi:10.1172/JCI128562 (2019).
- 6 Martin, C. J. *et al.* Selective inhibition of TGFbeta1 activation overcomes primary resistance to checkpoint blockade therapy by altering tumor immune landscape. *Sci Transl Med* **12**, doi:10.1126/scitranslmed.aay8456 (2020).
- 7 Cane, S., Van Snick, J., Uyttenhove, C., Pilotte, L. & Van den Eynde, B. J. TGFbeta1 neutralization displays therapeutic efficacy through both an immunomodulatory and a non-immune tumor-intrinsic mechanism. *J Immunother Cancer* **9**, doi:10.1136/jitc-2020-001798 (2021).
- 8 Terabe, M. *et al.* Blockade of only TGF-beta 1 and 2 is sufficient to enhance the efficacy of vaccine and PD-1 checkpoint blockade immunotherapy. *Oncoimmunology* **6**, e1308616, doi:10.1080/2162402X.2017.1308616 (2017).
- 9 Varadarajan, N. *et al.* A high-throughput single-cell analysis of human CD8(+) T cell functions reveals discordance for cytokine secretion and cytolysis. *J Clin Invest* **121**, 4322-4331, doi:10.1172/JCI58653 (2011).
- 10 Gunderson, A. J. *et al.* TGFbeta suppresses CD8(+) T cell expression of CXCR3 and tumor trafficking. *Nat Commun* **11**, 1749, doi:10.1038/s41467-020-15404-8 (2020).
- 11 Chen, M. L., Yan, B. S., Kozoriz, D. & Weiner, H. L. Novel CD8+ Treg suppress EAE by TGF-beta- and IFN-gamma-dependent mechanisms. *Eur J Immunol* **39**, 3423-3435, doi:10.1002/eji.200939441 (2009).

REVIEWERS' COMMENTS:

Reviewer #1 (Remarks to the Author):

I am pleased to see the authors have substantially and carefully addressed all the issues we concerned. The data organized in this version was more impressive and logical. This is an interesting work that potentially contributes to the clinical treatment.